# Stepwise progression of β-selection during T cell development involves histone deacetylation

Anchi S Chann[1,2,3,9] , Mirren Charnley[1,2] , Lucas M Newton[4] , Andrea Newbold[2,5] , Florian Wiede[3,9] , Tony Tiganis[3,9] , Patrick O Humbert[4,6,7,8] , Ricky W Johnstone[2,5] , Sarah M Russell[1,2,5]

**During T cell development, the first step in creating a unique T cell receptor (TCR) is genetic recombination of the TCRβ chain. The quality of the new TCRβ is assessed at the β-selection checkpoint. Most cells fail this checkpoint and die, but the coordination of fate at the β-selection checkpoint is not yet understood. We shed new light on fate determination during β-selection using a selective inhibitor of histone deacetylase 6, ACY1215. ACY1215 disrupted the β-selection checkpoint. Characterising the basis for this disruption revealed a new, pivotal stage in β-selection, bookended by up-regulation of TCR co-receptors, CD28 and CD2, respectively. Within this "DN3b^Pre" stage, CD5 and Lef1 are up-regulated to reflect pre-TCR signalling, and their expression correlates with proliferation. These findings suggest a refined model of β-selection in which a coordinated increase in expression of pre-TCR, CD28, CD5 and Lef1 allows for modulating TCR signalling strength and culminates in the expression of CD2 to enable exit from the β-selection checkpoint.**

## Introduction

T cells are generated in a multistep, highly orchestrated process guided by stromal cells in the thymus. T cell development involves massive cell proliferation interspersed with genomic recombination of the T cell receptor (TCR). If genomic recombination, proliferation, and death are not appropriately orchestrated, the implications for cancer and immunity are profound. Uncontrolled self-renewal or a sub-optimal TCR repertoire can lead to T cell acute lymphoblastic lymphoma, autoimmunity, or limited response to pathogens or cancers (Douek et al, 2000; Ballesteros-Arias et al, 2019; Notarangelo et al, 2020). T cell development is most tightly controlled during genomic recombination to generate the TCR. For

the most common αβ T cells, the β chain is generated at the DN3a phase of T cell development (see Fig S1A for a description of T cell development and the surface markers by which it is assessed), and the appropriateness of this TCRβ sequence is assessed during the β-selection checkpoint. During β-selection, the strength of the TCRβ chain is tested in combination with a surrogate "pre-TCR" before the TCRα chain is created (Chann & Russell, 2019; Dutta et al, 2021).

β-selection is in part determined by stromal cells in the thymic microenvironment, which provide Notch and chemokine signals that promote survival and differentiation (Chann & Russell, 2019; Dutta et al, 2021). Recently, it has become clear that interactions between the pre-TCR and the MHC-peptide on stromal cells also provide an opportunity to test TCR signalling (Allam et al, 2021; Dutta et al, 2021; Li et al, 2021) These interactions trigger a sequence of transcriptional changes and epigenetic modifications that ensure only the cells with an appropriately recombined TCRβ chain survive, proliferate, and differentiate. How pre-TCR signalling integrates fate determination including proliferation, differentiation, and survival during β-selection is still not understood (Chann & Russell, 2019; Dutta et al, 2021).

Epigenetic therapies such as small-molecule inhibitors of histone deacetylases (HDAC) have shown great promise in mediating clinical antitumor responses and are emerging as immune modulators (Hogg et al, 2020). The HDAC6-selective inhibitor ACY1215 (ricolinostat) can synergize with the BET inhibitor, JQ1, to recruit T cell–mediated responses to lung adenocarcinoma, enhance T cell responses to ex vivo melanoma cultures, impair effector CD8 T cell function during skin inflammation, and alter Treg function (Tsuji et al, 2015; Adeegbe et al, 2017; Xu et al, 2018; Laino et al, 2019; Zhang et al, 2021). HDACs 1, 2, and 3 are required at multiple stages of T cell development, but a role for HDAC6 in T cell development has not been identified (Wang et al, 2020). Mice lacking HDAC6 showed no overt T cell phenotypes (Zhang et al, 2008; Shapiro & Shapiro, 2020), and to our knowledge, the effect of acute deletion of HDAC6 or pharmacological inhibition of HDAC6 on T cell development has not

[1]Optical Sciences Centre, School of Science, Swinburne University of Technology, Hawthorn, Australia   [2]Peter MacCallum Cancer Centre, Melbourne, Australia   [3]Monash Biomedicine Discovery Institute, Monash University, Clayton, Australia   [4]Department of Biochemistry and Chemistry, La Trobe Institute for Molecular Science, La Trobe University, Melbourne, Australia   [5]Sir Peter MacCallum Department of Oncology, The University of Melbourne, Melbourne, Australia   [6]Research Centre for Molecular Cancer Prevention, La Trobe University, Melbourne, Australia   [7]Department of Biochemistry and Pharmacology, University of Melbourne, Melbourne, Australia   [8]Department of Clinical Pathology, University of Melbourne, Melbourne, Australia   [9]Department of Biochemistry and Molecular Biology, Monash University, Clayton, Australia

Correspondence: sarah.russell@petermac.org

been reported (Shapiro & Shapiro, 2020). Here, we show a striking effect of treatment with the HDAC6-selective inhibitor, ACY1215, on β-selection.

This effect of ACY1215 allowed us to define several stages of β-selection with high resolution. After the DN3a stage marked by up-regulation of the co-receptor CD28, we observed a transitional stage marked by low levels of the co-receptor CD2 that we termed DN3b^Pre. DN3b^Pre cells are enriched by ACY1215 treatment. An up-regulation of CD2 marks passage through the β-selection checkpoint to a phase we termed DN3b^Post. The transcription factor Lef1 is up-regulated in DN3b^Pre cells, together with an increasing expression of the reporter of TCR signalling, CD5. These findings implicate a refined model of β-selection in which, after expression of CD27 and CD28, a coordinated increase in expression of pre-TCR, CD5, and Lef1 provides for an escalating test of TCR signalling. Expression of CD2 marks the passing of this test and exit from the β-selection checkpoint.

## Results

### Treatment with ACY1215 impairs T cell differentiation at the β-selection stage

To examine the effects of ACY1215 on T cell development (Fig S1B and C), we used two models. Firstly, a well-validated in vitro model in which hematopoietic stem cells (HSCs) extracted from mouse fetal liver were cocultured with the OP9 mouse bone marrow stromal cell line transfected with the Notch ligand, DL1 ("OP9-DL1") (Holmes & Zúñiga-Pflücker, 2009). To assess relevance of our findings to physiological T cell development in the thymus, we also analysed cells directly taken from young adult mice.

At day 8–10 of the co-culture, when DN cells from mouse fetal liver were predominantly at the DN3 stage of T cell development (Fig S1B), we applied the HDAC6 inhibitor, ACY1215, at a concentration previously reported to alter function of mature T cells without affecting lymphocyte viability (Laino et al, 2019). Compared with DMSO-treated cells, cells treated with ACY1215 for 2 d (Fig 1A) showed reduced proportions of the later stages of differentiation, suggesting HDAC6 is required for optimal T cell development to the DP stage. The impact was directly on the T cells because ACY1215 had no effect on growth or acetylation of tubulin in the OP9-DL1 stromal cells (Fig S2A and B), and ACY1215-treated stromal cells provided proper support for thymocyte attachment and differentiation (Fig S2B and C). The shift in differentiation caused by ACY1215 was accompanied by a generalised inhibition of expansion in cell numbers at several stages of T cell development (Fig S2D). Comparing proportions at different DN stages using CD25 and CD44 expression (Fig 1A) showed an increase in the proportions of DN3 cells and commensurate decrease in proportions of DN4, suggesting that the ACY1215-treated cells were blocked at the DN3 stage.

The DN3 stage incorporates the β-selection checkpoint. The passing of the β-selection checkpoint, conventionally conceptualised as DN3a-to-DN3b progression, is marked by the expression of CD28 (Williams et al, 2005; Teague et al, 2010). Therefore,

we assessed whether DN3a (DN3, CD28^Lo) cells were inhibited from progressing to DN3b (DN3, CD28^Hi) by ACY1215 treatment over 1 and 2 d (Fig 1B). Surprisingly, the reverse was observed. Co-staining with CD25 and CD28 indicated a relatively stable distribution between DN3a and DN3b at 1 and 2 d in the control cultures but increasingly greater proportions of DN3b cells in the ACY1215-treated cultures. A similar shift to DN3b and DN4 at the expense of DN3a was observed in thymocytes extracted from an adult mouse and cultured for 1 d on OP9-DL1, although the relatively low number of DN3a in primary thymocytes ex vivo (Fig S1C) made the shift less obvious (Fig 1C).

We explored the cellular basis of this shift away from DN3a towards DN3b. Cell counts of cultures of sorted DN3a cells showed that the DN3a cells expanded more than fourfold over 2 d of control DMSO-treated culture, but this was dramatically reduced by ACY1215 treatment (Fig 1B). In contrast, ACY1215 caused a negligible reduction in absolute numbers of DN3b and DN4 cells arising from DN3a. This stage-specific effect was probably not due to differences in proliferation or death because CFSE and Ki-67 labelling showed similar inhibition of proliferation of DN3a and DN3b cells (Fig S3), and neither DN3a or DN3b cells were demonstrably susceptible to cell death, although DP cells died more in the presence of ACY1215 (Fig S4). Together, these data suggest that ACY1215 treatment of DN3a cells led to reduced amplification but little impact or a compensatory increase in the number of cells passing through the DN3b stage.

To reconcile the seemingly normal progression through β-selection (as indicated by DN3b and DN4 counts) with the block in DP cells, we first monitored differentiation from sorted DN3a and DN3b cells that had been treated for 4 d with ACY1215 or DMSO (Fig S5). Progression past the DN3 stages was inhibited for both DN3a and DN3b starting populations, although less clear for the DN3a starting population because of limited differentiation by day 4. To confirm that the effect at β-selection led to a downstream impact on DP cells, we counted the number of cells at each differentiation stage that developed from the sorted DN3b population (Fig 1D). By day 4, DN3b cells had expanded more than 50-fold, with a negligible reduction caused by treatment with ACY1215. DN4 cell numbers were also only slightly reduced by ACY1215 treatment, but DP were reduced by over two orders of magnitude. These data together suggest that ACY1215 treatment impacts generally upon cell expansion but also alters differentiation to DP by acting at the DN3 stage, perhaps by causing precocious differentiation from DN3a to DN3b.

The DN3 cell stage involves fate choice between two T cell lineages (αβ cells and γδ cells), each expressing a different set of TCRs (Hayday & Pennington, 2007). To determine which lineage was altered by ACY1215, we cultured developing T cells for 2 d with and without ACY1215 and labelled the cells with antibodies specific for these TCRs to assess progression through the DN and DP stages (Fig S6). ACY1215 treatment reduced the proportion of TCRβ-expressing cells at all stages after β-selection but did not inhibit TCRγδ development. Together, these data indicated that the defect in differentiation to DP after ACY1215 treatment was specific to αβ T cells and likely involved the β-selection checkpoint.

One possible explanation for this passage to DN3b without optimal subsequent differentiation could be that ACY1215-treated cells express some markers suggestive of differentiation but are not

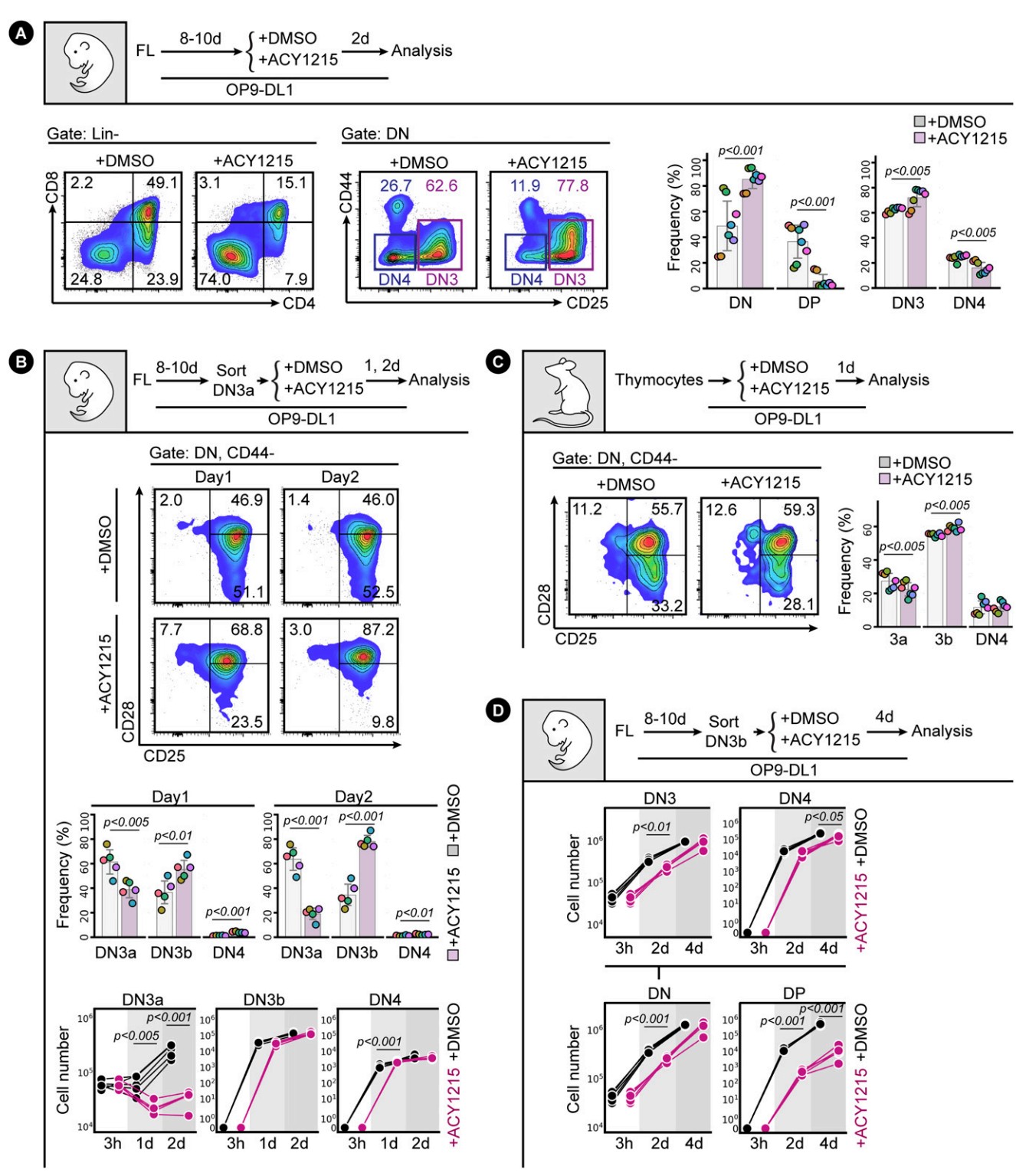

**Figure 1. ACY1215 inhibits T cell development at the DN3 stage.**
**(A)** FL-derived developing T cells were treated with ACY1215 or DMSO for 2 d as indicated, and the Lin⁻ cells were assessed for CD4 and CD8 expression to determine the proportions of DN and DP populations. The DN (Lin⁻, CD4⁻, CD8⁻) cells were assessed for CD44 and CD25 expression to determine the proportions of DN3 and DN4 populations. The frequency of each population was summarised in a swarm plot, with individual colours representing each of eight experiments. **(B, C)** Sorted, FL-derived DN3a cells (B) or thymocytes (C) were cultured under ACY1215 or DMSO treatment for 1 or 2 d as indicated, and the CD44⁻ DN population (representing DN3 and DN4 stages) was assessed for CD25 and CD28 expression. The frequency of each population was summarised in swarm plots, with individual colours representing each of five or seven experiments, respectively. Cell numbers were summarised in line plots, summarising the same number of experiments. **(D)** Sorted, FL-derived DN3b cells were

competent precursors of $\alpha\beta$ T cells. To assess this, we first characterised several phenotypic markers implicated in progression through the $\beta$-selection checkpoint. Co-staining DN3 cells with both CD28 and CD27, both of which can discriminate between DN3a and DN3b cells (Taghon et al, 2006; Teague et al, 2010), clearly discriminated two subpopulations in DN3 cells pre-treated with DMSO (Fig 2A). A shift from DN3a to DN3b is observed for both fetal liver–derived lymphocytes and ex vivo extracted thymocytes cultured on OP9-DL1 stroma. These data confirm the findings above that, at least by the canonical markers, CD27 and CD28, ACY1215 treatment promotes differentiation towards DN3b at the expense of DN3a.

Given that the primary function of this checkpoint is to ensure appropriate TCR$\beta$ rearrangement, we assessed the correlation of each marker with TCR$\beta$ expression (Fig 2B). As previously shown (Williams et al, 2005), expression of CD28 was clearly correlated with expression of icTCR$\beta$ in cells derived from fetal liver or from the adult thymus. The CD28$^{Hi}$ cells also expressed more of the nutrient transporters, CD71 and CD98, and more surface TCR$\beta$ (Fig S7A) indicating that, in general, they had progressed further through $\beta$-selection, but the correlation was not absolute. For both cell sources, ACY1215 treatment resulted in less icTCR$\beta$ expression in the DN3b (whether measured as CD28$^{Hi}$ or CD27$^{Hi}$ DN3 cells) (Fig 2B, see also Fig S8C). To validate the icTCR$\beta$ staining, we directly compared it with surface TCR$\beta$ (Fig S7B). For both cell sources, expression of surface TCR$\beta$ was highly correlated with that of icTCR$\beta$ and both were reduced by ACY1215 treatment. Thus, besides depleting DN3a cells, treatment with ACY1215 increases the proportions of a cell population that bears many of the phenotypic markers of passage through $\beta$-selection, but these cells appear somewhat defective in expression of TCR$\beta$ and do not progress optimally through the DP stage. We speculated that further characterisation of these cells might yield valuable information about requirements at the $\beta$-selection checkpoint.

### CD2 expression identifies DN3b$^{Pre}$ and DN3b$^{Post}$ as two functionally distinct stages in $\beta$-selection

A well-established co-stimulator of TCR signalling is CD2 (Skånland et al, 2014). Expression of CD2 marks TCR$\beta$ V(D)J rearrangement and can facilitate pre-TCR signalling for passage through the $\beta$-selection checkpoint (Groettrup et al, 1992; Rodewald et al, 1993; Sasada & Reinherz, 2001). In the next series of experiments, we find that CD28 and CD2 reliably define three sequential stages of $\beta$ selection in DN3 cells, which we term DN3a, DN3b$^{Pre}$, and DN3b$^{Post}$ to accommodate pre-existing nomenclature (Fig 3A). Co-staining of DMSO-treated DN3 cells with CD28 and CD2 showed that all DN3a cells (CD28$^{Lo}$) were CD2$^{Lo}$, but DN3b (CD28$^{Hi}$) cells had two clearly discernible populations of CD2$^{Lo}$ (DN3b$^{Pre}$) and CD2$^{Hi}$ (DN3b$^{Post}$) (Fig 3Bi). ACY1215 treatment enriched for the DN3b$^{Pre}$. Cell counts indicated that the number of DN3b$^{Pre}$ cells was not substantially affected by ACY1215, but DN3a and DN3b$^{Post}$ numbers were dramatically reduced (Fig S8A). Thus, ACY1215 treatment led to an

enrichment in a DN3b$^{Pre}$ population that was low for CD2. The DN3b$^{Pre}$ population (Fig 3Bii) was evident in thymocytes from young adult mice, even without in vitro culture (see Fig 4 below) and was again enriched by treatment with ACY1215.

To confirm that DN3b$^{Pre}$ cells were earlier in $\beta$-selection than DN3b$^{Post}$ cells, we assessed the association of CD2 expression with proliferation as measured by CFSE dilution (Fig S8B). Two clear populations were evident, with the CD2$^{Hi}$ (DN3b$^{Post}$) cells having undergone substantially more proliferation than the CD2$^{Lo}$ (DN3b$^{Pre}$) cells. ACY1215 treatment did not disrupt the association between CD2 and proliferation. Expression of CD27 and surface TCR$\beta$ was also increased from DN3a, DN3b$^{Pre}$, to DN3b$^{Post}$, with stepwise correlation which was not altered by ACY1215 despite lower surface TCR$\beta$ on DN3b$^{Post}$ cells (Fig S8C). We then assessed icTCR$\beta$ and Ki-67 staining, which should reflect the cell proliferation triggered by successful passage through the $\beta$-selection checkpoint (Fig 3C). Indeed, consistent with previous findings that TCR$\beta$ V(D)J recombination is restricted to DN cells positive for CD2 (Rodewald et al, 1993), DN3b$^{Pre}$ cells were intermediate in expression of icTCR$\beta$ and Ki-67 between the levels seen on DN3a cells and DN3b$^{Post}$ cells. The correlated, stepwise increase in icTCR$\beta$ and Ki-67 was observed in cultures from both fetal liver and adult thymus (Fig 3Ci and ii). This pattern was even more striking in thymocytes from adult mice, where DN3a cells showed no proliferation, DN3b$^{Pre}$ cells were heterogeneous with respect to proliferation, and proliferation correlated with icTCR$\beta$ expression, and DN3b$^{Post}$ cells were all highly proliferative, with minimal impact from ACY1215 treatment. These data suggest that the dramatic reduction in TCR$\beta$ expression upon ACY1215 treatment observed in Fig 2 was explained more by the enrichment of DN3b$^{Pre}$ cells than a direct effect on TCR$\beta$ expression. c-Myc, which mediates the proliferative response to pre-TCR signalling but is not required for differentiation (Dose et al, 2006), showed a similar stepwise increase with these differentiation stages (Fig S8D). These data are all compatible with an interim DN3b$^{Pre}$ stage that reflects an early and pivotal step in $\beta$-selection where rearrangement of the TCR$\beta$ chain is translated into proliferation.

To formally assess whether the DN3b$^{Pre}$ stage is a transitional stage before DN3b$^{Post}$, we sorted DN3a, DN3b$^{Pre}$, and DN3b$^{Post}$ cells from adult mice and cultured them for further analysis (Fig 4). Analysis of the cells immediately after extraction (top right contour plots) showed clearly distinct DN3a, DN3b$^{Pre}$, and DN3b$^{Post}$ cells. After 2 d, DN3b$^{Post}$ cells had produced substantial number of DP cells, with fewer produced by DN3b$^{Pre}$ and fewer still by DN3a cells. Also, compatible with unidirectional progression from DN3a to DN3b$^{Pre}$ to DN3b$^{Post}$ was the proportion of DN4 and DN3 stages from each sorted population. ACY1215 treatment of thymocytes stabilized the DN3b$^{Pre}$ population and blocked progression to DP. Similar data were obtained in the fetal liver–derived cultures (Fig S9). These data confirm that DN3b$^{Pre}$ (CD28$^{Hi}$CD2$^{Lo}$) is a transitional state between DN3a (CD28$^{Lo}$CD2$^{Lo}$) and DN3b$^{Post}$ (CD28$^{Hi}$CD2$^{Hi}$), reflecting partial progression through $\beta$-selection.

cultured for 4 d under ACY1215 or DMSO treatment as indicated, and cell numbers per population were estimated by multiplying the total cell number by the frequency, and five experiments were summarised in the line chart.

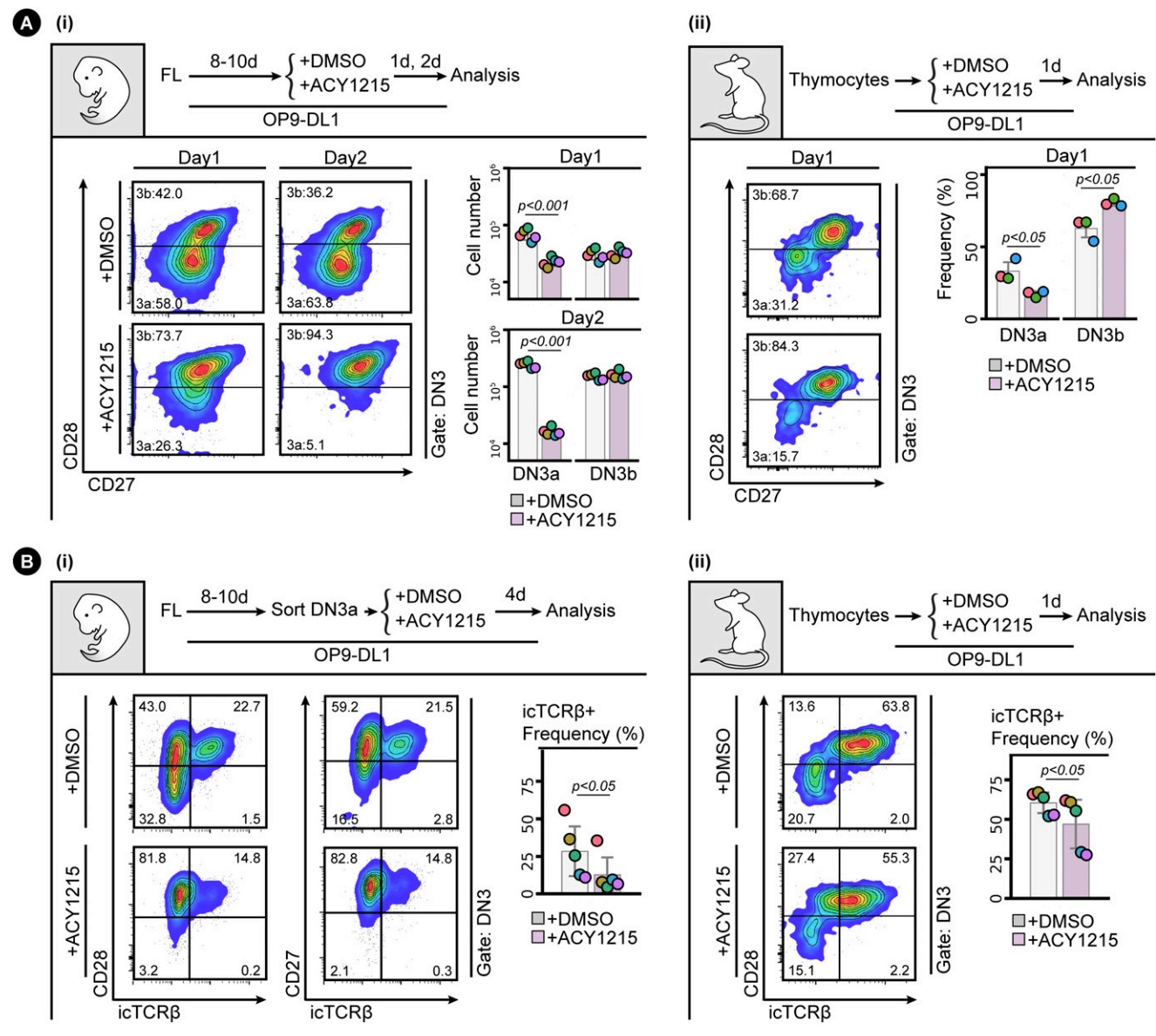

**Figure 2. ACY1215 inhibits β-selection.**
**(A)** Developing T cells from FL (i) or the adult thymus (ii) were treated with ACY1215 or DMSO for 1 or 2 d, and the DN3 cells were assessed for CD27 and CD28 expression to discriminate DN3a and DN3b. Cell number or frequency was summarised in the swarm plot, with individual colours representing each of five and three experiments, respectively. **(B)** Sorted, FL-derived DN3a cells (i) were cultured under ACY1215 or DMSO treatment for 4 d as indicated, and DN3 cells were assessed for the expressions of intracellular TCRβ (icTCRβ) versus CD28 or CD27. Thymocytes (ii) were cultured under ACY1215 or DMSO treatment for 1 d as indicated, and DN3 cells were assessed for the expressions of icTCRβ versus CD28. The frequency of icTCRβ⁺ cells was summarised in a swarm plot, with individual colours representing each of five experiments.

## ACY1215 alters the pre-TCR signalling platform clustered at microtubule-organising centre (MTOC)

The enrichment of CD28^Hi^CD2^Lo^ (DN3b^Pre^) cells by ACY1215 treatment in both fetal liver–derived cells and thymocytes from adult mice suggests that a key process in this stage might involve acetylation. We explored three possible mechanisms by which ACY1215 might disrupt β-selection. ACY1215 is selective for HDAC6 and can inhibit the catalytic activity of HDAC6 (Eich et al, 2018; Narita et al, 2019). HDAC6 protein is expressed at every stage of T cell development (Fig S10A). The best known role of HDAC6 is in modification of histones

for epigenetic regulation. Indeed, acetylation of H3K18 was increased upon treatment with ACY1215, suggesting that epigenetic modification could be affected by the drug (Fig S10B). Although best known as a regulator of histone acetylation, HDAC6 can also acetylate α-tubulin (Narita et al, 2019). In mature T cells, the regulation of acetylation of α-tubulin by HDAC6 influences TCR-mediated recruitment of the MTOC to the immunological synapse (Serrador et al, 2004). In all T cell developmental stages, HDAC6 was predominantly in the nucleus but also detectable in the cytoplasm and near the MTOC, compatible with effects on MTOC modification (Fig S10C–E). Acetylation of α-tubulin (K40), which can be de-acetylated by HDAC6 (Osseni et al,

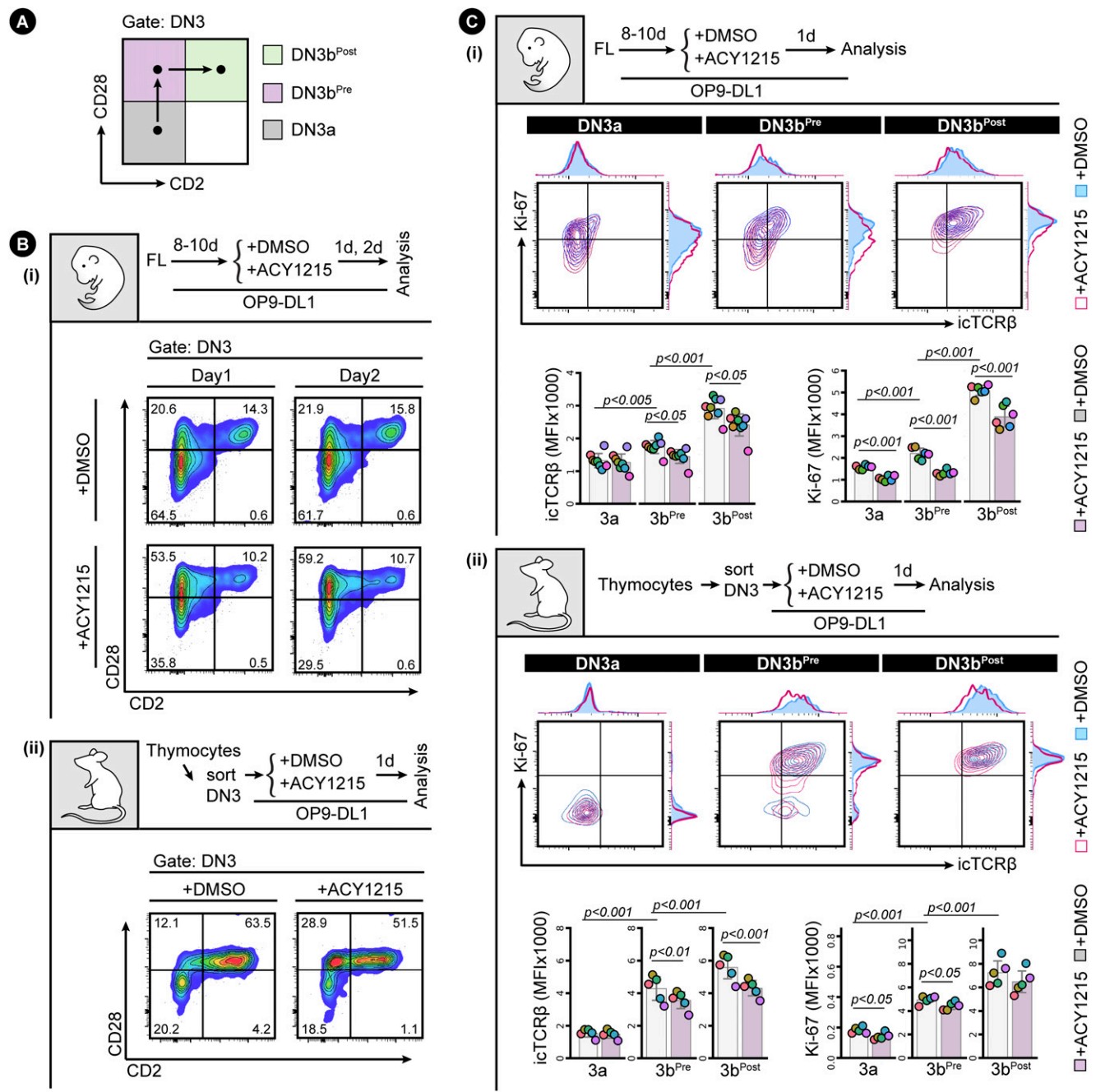

**Figure 3. CD2 up-regulation discriminates DN3b^Pre and DN3b^Post populations.**
**(A)** The strategy for discriminating between three populations (DN3a, DN3b^Pre, and DN3b^Post) is indicated in the schematic. **(B)** FL-derived developing T cells (i) or the sorted thymocyte DN3 (ii) were treated with ACY1215 or DMSO for 1 or 2 d, and the DN3 cells were assessed for CD2 and CD28 expression. **(C)** FL-derived developing T cells (i) or the sorted thymus–derived DN3 (ii) were treated with ACY1215 or DMSO for 1 d, and DN3a, DN3b^Pre, and DN3b^Post cells were assessed for icTCRβ and Ki-67 expression. MFI of icTCRβ and Ki-67 were summarised in histogram and swarm plot, with individual colours representing each of 5~8 experiments.

2020), was evident throughout the T cell differentiation stages (Fig 5A). Upon a 2-h treatment with ACY1215, each population showed an increase in acetylated tubulin, suggesting HDAC6 de-acetylated tubulin throughout early T cell development.

We previously found that T cells undergoing β-selection form an immunological synapse in which MTOC and other TCR-associated signalling proteins move with the MTOC to the site of antigen

receptor signalling (Allam et al, 2021). We therefore speculated that an alteration to MTOC organisation might mediate the effects of ACY1215 on β-selection by impacting the immunological synapse. To address this, we first assessed the association of LAT with the MTOC (Fig 5B). DN3a cells were cultured for 1 d with and without ACY1215 (so would be predominantly DN3a and DN3b^Pre and not yet past β-selection; see Fig S9). Acetyl-α-tubulin presented as a single cluster in the site of the

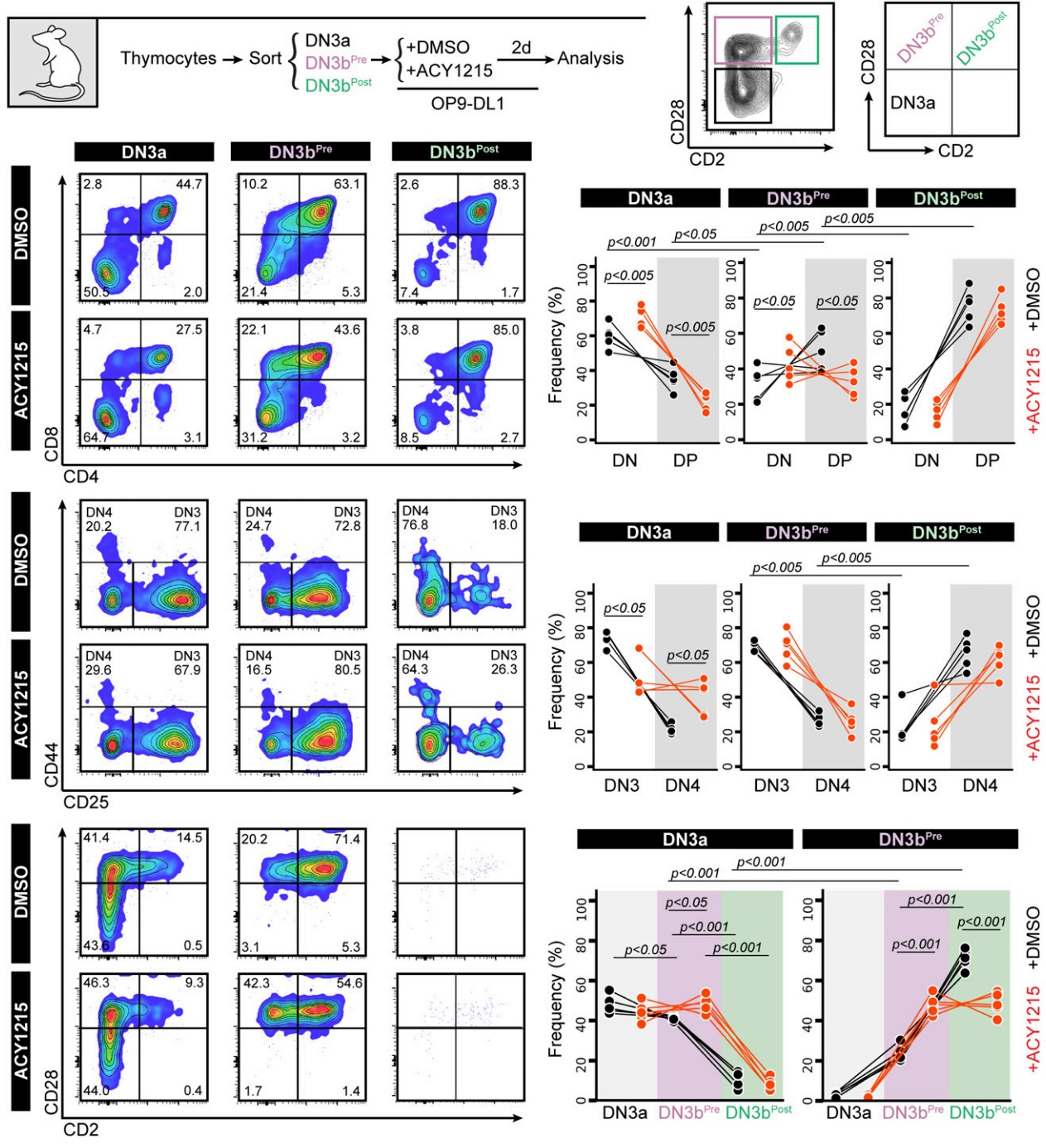

**Figure 4. DN3a, DN3b^Pre and DN3b^Post stages follow a sequential progression.**
The DN3a, DN3b^Pre, and DN3b^Post cells from the adult thymus were sorted as indicated and were cultured on OP9-DL1. After 2 d, the cells were analysed by CD4 versus CD8, the DN cells were analysed by CD44 versus CD25, and the DN3 cells were analysed by CD2 versus CD28. The frequencies of each population from five experiments were summarised in the line plots.

MTOC and was increased in intensity with ACY1215 treatment, compatible with the flow cytometry results. LAT was strongly associated with the MTOC in DMSO-treated cells but altered by ACY1215 treatment, with less intense fluorescence and less clustering at the MTOC. Notch is particularly important during β-selection in mediating assembly of the immunological synapse and transmitting signals downstream of

the pre-TCR signal (Charnley et al, 2020; Allam et al, 2021). We therefore assessed the co-localisation of Notch1 and the MTOC in DN3a cells cultured for 1 d with and without ACY1215 (Fig 5C). In DMSO-treated cells, Notch was concentrated at the MTOC but, similar to LAT, was much more diffuse after ACY1215 treatment. Interestingly, the change in localisation of both Notch and LAT

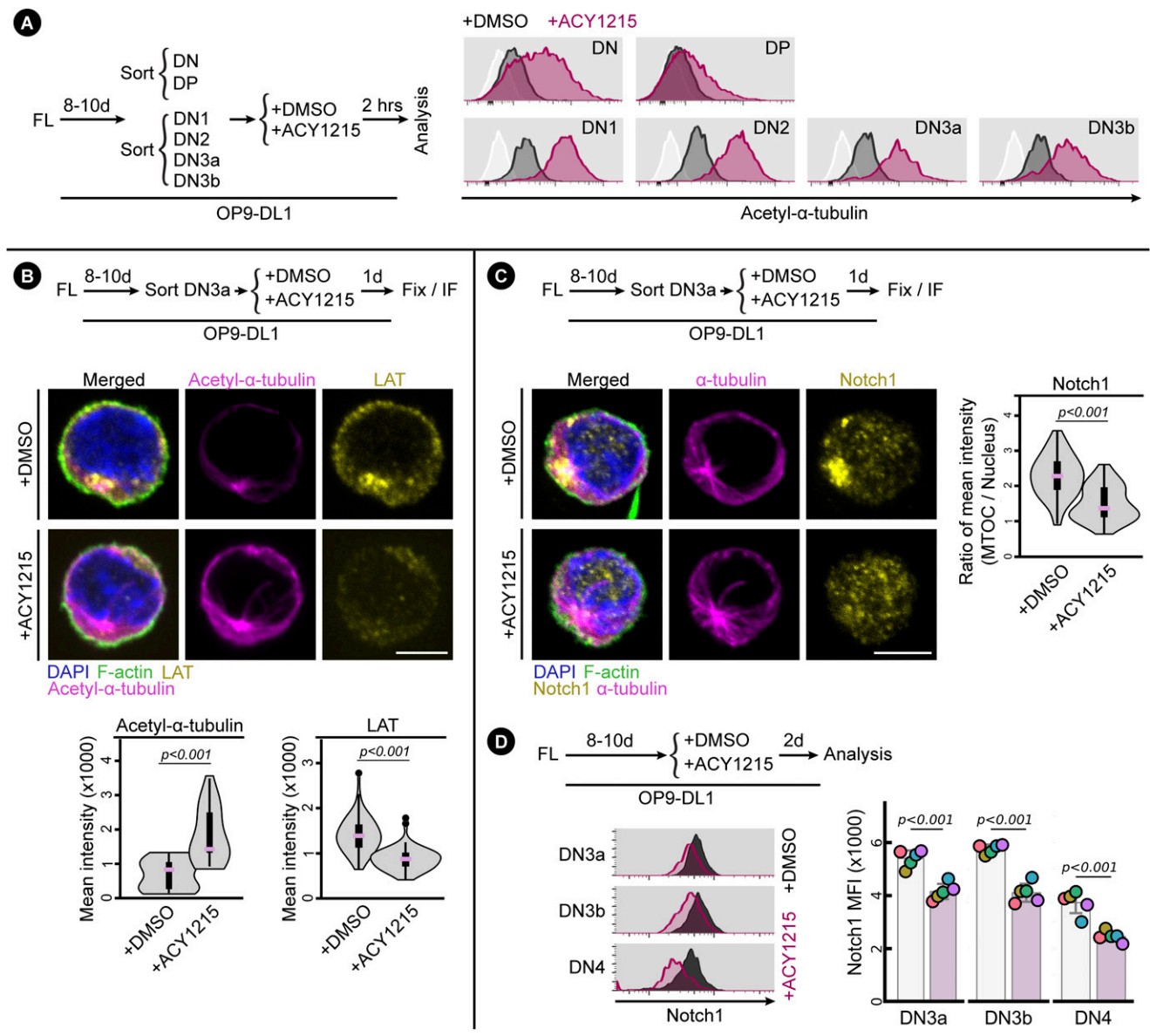

**Figure 5. ACY1215 alters pre-TCR signalling clustered at the MTOC.**
**(A)** Sorted FL-derived DN, DP, DN1, DN2, DN3a, DN3b were treated with ACY1215 or DMSO for 2 h as indicated and were assessed for acetyl-α-tubulin (K40) as shown in histograms. The white histograms are unstained controls in which cells from each sorting were stained with secondary antibody only. Similar results were observed in three separate experiments. **(B, C)** Sorted FL-derived DN3a cells were treated with ACY1215 or DMSO for 1 d as indicated, followed by immunostaining for LAT, acetyl-α-tubulin (K40), Notch1 and α-tubulin. The mean intensity of acetyl-α-tubulin, LAT, and the ratio of Notch1 expressed at MTOC versus at nucleus were summarised in violin plot overlaid with box plot. Three experiments were included. Cell number for LAT staining: ACY1215 (n = 55) and DMSO (n = 54); cell number for acetyl-α-tubulin (K40) staining: ACY1215 (n = 30) and DMSO (n = 31); cell number for Notch1 staining: ACY1215 (n = 56) and DMSO (n = 55). **(D)** FL-derived developing T cells were treated with ACY1215 or DMSO for 2 d, and the DN3a, DN3b, and DN4 cells were assessed for Notch1 expression, as summarised in swarm plot, with individual colours representing each of five experiments.

was accompanied by a loss in their intensity. A reduction in surface Notch expression was observed in DN3a, DN3b, and DN4 cells, suggesting that the reduction in Notch1 transcription that normally occurs after β-selection (Yashiro-Ohtani et al, 2009) might be accelerated by ACY1215 (Fig 5D). Together, these data suggest that a possible mechanistic basis for the effects of ACY1215 on β-selection is via alterations in recruitment by MTOC of Notch1 and TCR signalling components to the immunological synapse, associated with enhanced tubulin acetylation at the immunological synapse.

### DN3b^Pre cells have commenced but not completed β-selection and are controlled by sequential changes in key regulators of β-selection

How might such a disruption in the immunological synapse and TCR signalling alter progression from DN3b^Pre to DN3b^Post? A collaboration between pre-TCR and Wnt signalling is well established, with two core Wnt pathway transcriptional regulators: β-catenin and Lef1/TCF-1 essential to β-selection (Travis et al, 1991; Gounari et al, 2001; Staal & Clevers, 2005; Xu et al, 2009; López-Rodríguez et al,

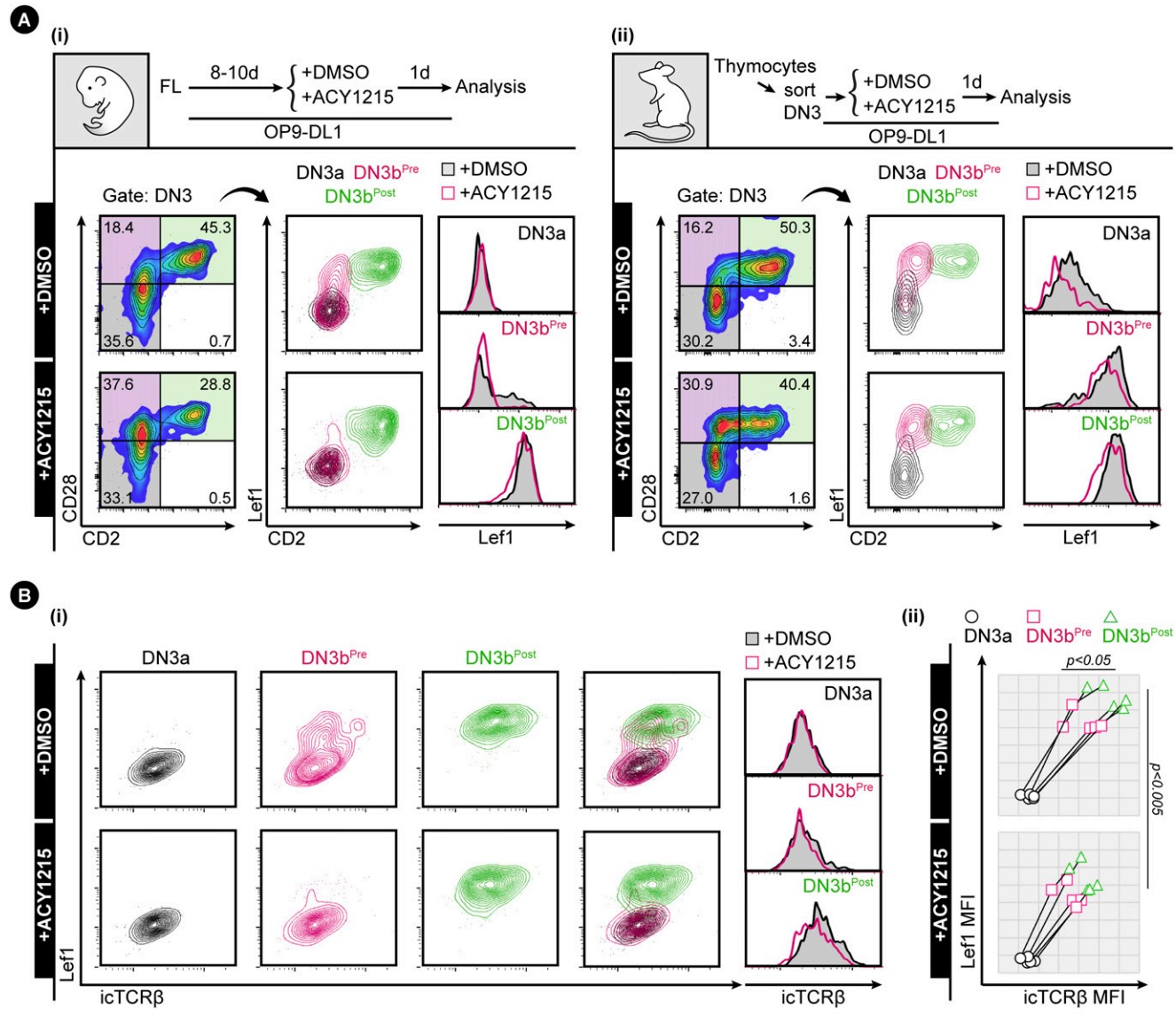

**Figure 6.  Up-regulation of Lef1 and TCRβ in DN3b^Pre cells.**
**(A)** FL-derived developing T cells (i) or the sorted thymocyte DN3 (ii) were treated with ACY1215 or DMSO for 1 d as indicated, and DN3 cells were assessed for CD2 and CD28 expression to gate for the DN3a, DN3b^Pre, and DN3b^Post populations. **(B)** DN3a, DN3b^Pre, and DN3b^Post cells from FL-derived developing T cells (i) or sorted thymocyte DN3 (ii) were assessed for Lef1 and icTCRβ expression. Quantification of Lef1 expression for (i) in six experiments is presented in Fig S11, of icTCRβ in five experiments in Fig S7, and the coordinated up-regulation of Lef1 and icTCRβ of five experiments for (ii) is shown in the line plot.

2015; Zhao et al, 2021). TCF-1 and Lef1 can directly modify histone acetylation, and this function is inhibited by the pharmacological HDAC inhibitors, tubacin and vorinostat (Xing et al, 2016; Zhao et al, 2021). In support of functional interplay between ACY1215 treatment and TCF-1 and Lef1, TCF-1 deficiency leads to a similar phenotype as ACY1215 treatment, with disrupted β-selection and reduced expression of icTCRβ in DN4 cells (Goux et al, 2005), and deletion of both TCF-1 and Lef1 leads to a complete block at DN4 (Yu et al, 2012). We focused on Lef1 because Lef1 is specifically expressed at the β-selection stage (Mingueneau et al, 2013). To assess whether Lef1 might mediate the β-selection decisions at the DN3b^Pre stage, we explored how its expression changed with differentiation and how it was impacted by ACY1215 treatment. Lef1 was minimally expressed in DN3a cells, uniformly

bright in DN3b^Post cells but heterogeneous in DN3b^Pre cells derived from either fetal liver or adult thymus (Figs 6Ai and ii and S11). In the DN3b^Pre cells, Lef1 expression was associated with increased expression of TCRβ, albeit with a wide range in TCRβ expression (Fig 6B). ACY1215 treatment did not demonstrably alter icTCRβ expression (Fig 6B) but prevented the up-regulation of Lef1 expression in DN3b^Pre cells, suggesting a role in pre-TCR-mediated induction of Lef1.

The correlation in expression of TCRβ with proliferation and Lef1 expression raised the possibility that the pre-TCR signal might be directly translated into phenotype during the DN3b^Pre stage. To explore this, we investigated the TCR-associated receptor, CD5. A role for CD5 has not been demonstrated during β-selection, but during positive selection, expression of CD5 reports the strength of

TCR signalling and tunes TCR signalling responsiveness, dampening the response of cells to TCR signalling via the NF-κB pathway (Tarakhovsky et al, 1995; Azzam et al, 1998, 2001; Voisinne et al, 2018). To explore the dynamics of CD5 expression during β-selection, we stained fetal liver–derived DN3 cells with CD2, Lef1, and CD5 (Fig S11). Gating for DN3b^Pre with low and high Lef1 expression showed strong correlation of CD5 and Lef1 expression. Interestingly, Lef1 up-regulation was retained in the presence of ACY1215, but the correlation with CD5 expression was lost. These data suggest a model in which CD5 expression is downstream of Lef1 and suggest the effects of ACY1215 on progression through β-selection might be mediated by disrupting the functional connection between Lef1 and CD5. Lef1 and CD5 expression were similarly correlated in DN3 cells extracted from the adult thymus, particularly in DN3b^Pre cells (Fig 7A).

To confirm this model, we made use of the fact that progression past DN3 is marked by down-regulation of CD25, and a transitional stage between DN3b and DN4 (sometimes termed DN3c) can be identified within the DN3 population (Ananias et al, 2008; Teague et al, 2010). We therefore assessed the relationship between CD2 and CD25 in the DN3b cells. DN3b^Pre population was still high for CD25, but DN3b^post included CD25^Lo cells (Fig S12A). These data are compatible with the notion that DN3b^Post marks a later stage in differentiation, which includes, but is not exclusive to, the DN3c population (Ananias et al, 2008). To assess whether the combined expression of CD5 and Lef1 in DN3b cells indicated progression through β-selection, we gated for the two clearly discriminated subpopulations, Lef1^LoCD5^Lo and Lef1^HiCD5^Hi, and compared their CD25 expression (Fig S12B). The Lef1^HiCD5^Hi cells were substantially lower for CD25, indicating that they had already progressed through β-selection. Although as expected, there were far fewer cells in the Lef1^HiCD5^Hi population upon ACY1215 treatment, they also exhibited the reduction in CD25, suggesting that ACY1215-treated cells can precociously differentiate beyond β-selection in the absence of CD5 expression.

To confirm the relationship between pre-TCR and CD5 and to assess the impact of ACY121, we cultured DN3a or DN3b cells from adult thymus in the presence of ACY1215 or DMSO for 2 d, gated for DN3b cells, and compared CD5 expression to surface TCRβ (Fig 7B). Cells from the DN3a starting population were more likely to be dull for both CD5 and surface TCRβ compared with the more mature cells from the DN3b starting population. However, in these mature cells, expression of surface TCRβ correlated with expression of CD5, suggesting that, as in other cell types, CD5 expression might report TCR signalling strength during β-selection. In ACY1215-treated cells, expression of surface TCRβ was reduced by the DN3b^Post stage and uncoupled from CD5. Similar findings were obtained using fetal liver cultures and staining for CD5 and icTCRβ (Fig S13). The correlation of TCRβ with CD5, and its uncoupling after ACY1215 treatment, was not observed with CD27 and CD28 (Fig 2) nor CD2 (Fig 3). Together, a functional link between pre-TCR, Lef1, and CD5 is coordinated at DN3b^Pre stage (Figs 8A and S14A) and is affected by ACY1215.

To assess the functional relationship between TCRβ and CD5, we compared CD5 expression and proliferation in the presence and absence of ACY1215 (Fig S14B). Proliferation was minimal in most DN3a cells and DN3b^Pre cells but was evident in a portion of DN3b^Pre cells from the DN3b starting population and in DN3b^Post cells. This proliferation was clearly correlated with CD5 expression in control

cells. However, the correlation between CD5 expression and CFSE dilution was lost in the DN3b^Pre cells that had been treated with ACY1215. These data together indicate that CD5 is functionally connected to both TCRβ expression and proliferation, and that this functional association is broken by ACY1215 treatment.

# Discussion

A critical factor in β-selection is the sequential expression of specific transcriptional regulators, with differential perdurance and cross-inhibition ensuring checks and balances that prevent inappropriate survival and differentiation (Chann & Russell, 2019; Zhao et al, 2021). Many such checks and balances have been identified to play important roles during β-selection, but the stepwise progression of events that dictate fate has been elusive (Chann & Russell, 2019; Zhao et al, 2021). Our explorations into effects of an HDAC6 inhibitor have revealed a new working model (Fig 8B), Once DN3b^Pre cells have entered the β-selection checkpoint, they respond to graded levels of TCRβ by expressing Lef1, then CD5. Normally, proliferation only occurs in cells expressing CD5. Passing of the β-selection checkpoint is then marked by the expression of CD2. The disruption in TCRβ-associated expression of CD5 by ACY1215 provides a possible mechanism for the escape of cells lacking an appropriately rearranged TCRβ chain after ACY1215 treatment, although such a role remains to be investigated.

The up-regulation of CD27 and CD28 is gradual, but the up-regulation of CD2 is more abrupt, making for a convenient demarcation of differentiation stages. Our findings are compatible with previous observations that CD2 expression marks cells with TCRβ gene rearrangement (Rodewald et al, 1993) and that knockout of CD2 inhibits progression to the DN4 stage but does not influence TCRβ repertoire (Sasada & Reinherz, 2001) and suggestions of cis-regulation of CD2 and TCRβ expression involving chromatin structure (Kamoun et al, 1995). Together, these findings suggest that expression of CD2 (marking entrance to the DN3b^Post stage) reflects the completion of TCRβ recombination and triggers critical fate-determining events that involve progressive up-regulation of CD5 and Lef1. CD2 in mature T cells sets quantitative thresholds for activation (Bachmann et al, 1999), suggesting the possibility that a similar process could be involved in β-selection.

## Ideas and speculation

A novel finding from this work is that CD5 levels are modified in association with pre-TCR expression and that treatment with ACY1215 prevents the up-regulation of CD5 (but has a lesser effect on Lef1 up-regulation) during β-selection. At other stages of T cell development and activation, CD5 reports on the strength of TCR signalling, with increasing expression reflecting a stronger TCR signal (Tarakhovsky et al, 1995; Azzam et al, 1998; Sood et al, 2021). Our findings that the correlation of CD5 with TCRβ expression is lost with ACY1215 treatment, but the correlation of CD5 expression with proliferation is retained with ACY1215 treatment, suggest the possibility that CD5 plays a similar role as a reporter of pre-TCR

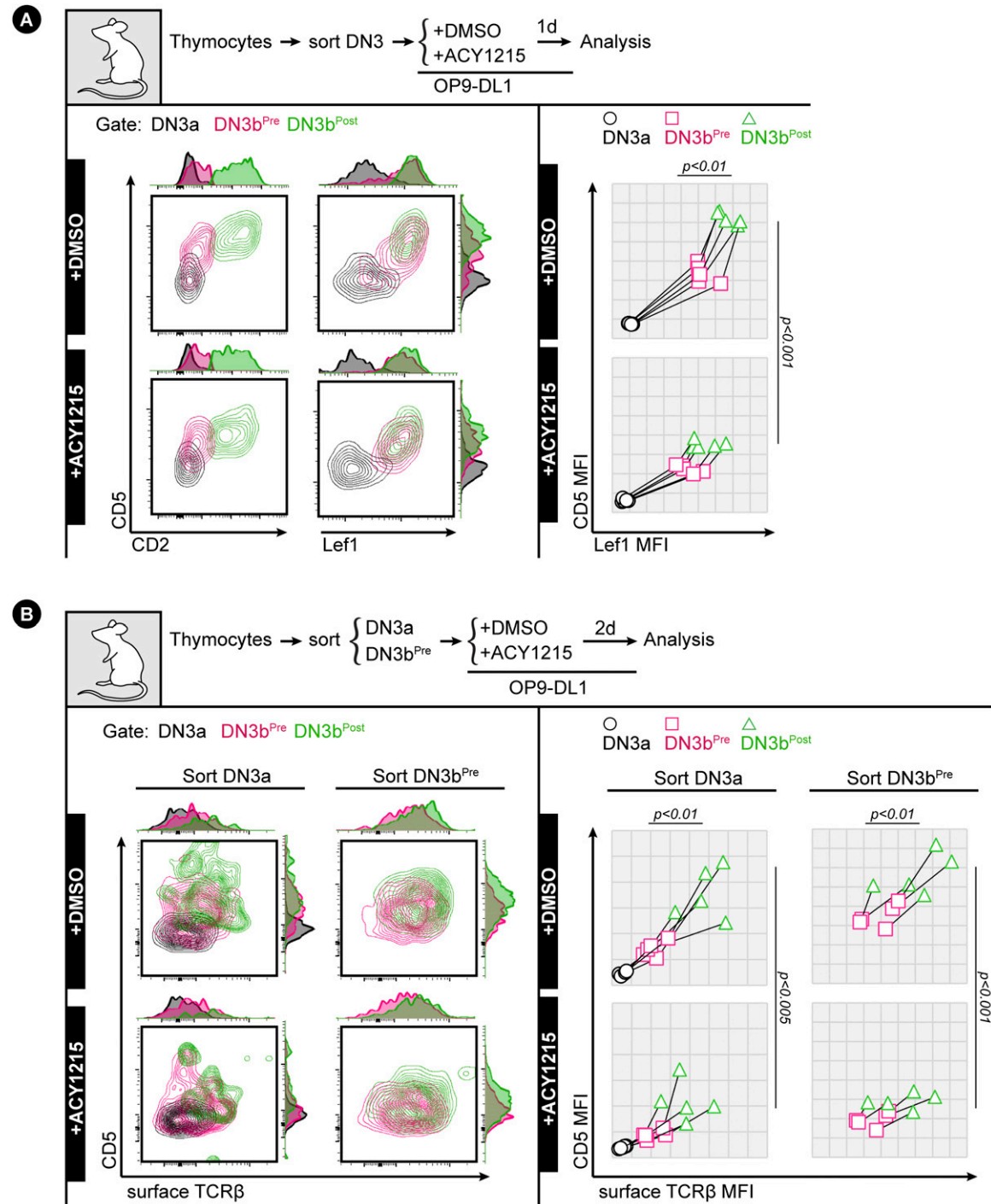

**Figure 7. CD5 is up-regulated with Lef1 and TCRβ in DN3b^Post cells.**
**(A)** Sorted DN3 cells from the adult thymus were treated with ACY1215 or DMSO for 1 d as indicated, and DN3 cells were assessed for CD2 and CD28 expression to gate for the DN3a, DN3b^Pre, and DN3b^Post populations, then analysed for CD5 and Lef1 expression. Data from five experiments were summarised in a line plot (right) or in swarm plots (shown in Fig S11). **(B)** Sorted DN3a and DN3b^Pre cells from the adult thymus were treated with ACY1215 or DMSO for 2 d as indicated, assessed for CD2 and CD28 expression to gate for the DN3a, DN3b^Pre, and DN3b^Post populations, then analysed for CD5 and surface TCRβ expression. Data from five experiments were summarised in a line plot (right) or in swarm plots (shown in Fig S11).

signalling. Interestingly, it is now emerging that differential CD5 expression also marks the propensity for different fates in thymocytes and CD4⁺ T cells, and this reflects transcriptional and epigenetic differences between cells with high and low CD5 expression (Rogers et al, 2021). In other cell types, the combined action of responding to, and dampening, the TCR signal allows CD5

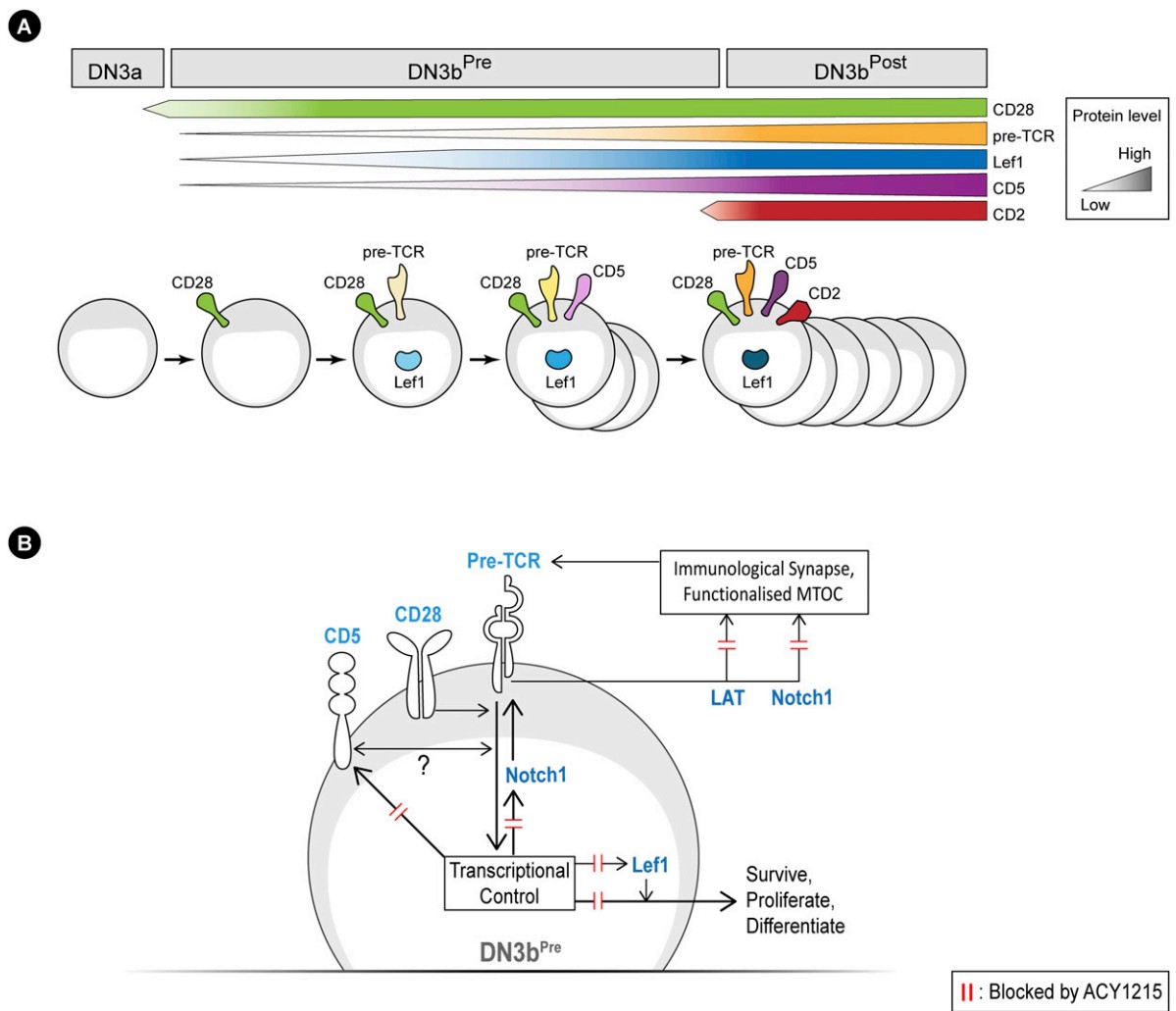

**Figure 8. TCR co-receptor up-regulation correlates with the passing of β-selection checkpoint.**
**(A)** Schematic describes the stepwise expression of TCR co-receptors (CD28, CD5, and CD2) and Lef1 from DN3a, DN3bPre to DN3bPost stage. **(B)** Schematic summarises the working model of molecular interactions at the DN3bPre stage. First, CD28 (the defining step in transition from DN3a to DN3bPre) acts to promote signalling through the nascent pre-TCR. Second, pre-TCR signals induce expression of CD5 (or vice versa) and lead to the passing of the β-selection checkpoint by coordinating transcriptional control, functional MTOC and immunological synapse formation for survival, proliferation, and differentiation. Possible molecular events disrupted by ACY1215 treatment to cause these cellular responses are indicated by the red double lines.

to tune the TCR signal and to shift the window and modify the timing of positive selection (Voisinne et al, 2018; Matson et al, 2020; Lutes et al, 2021; Sood et al, 2021). Whether or not such a tuning role might occur for CD5 during β-selection needs further exploration, but a possible role for tuning of the TCRβ repertoire during β selection is made relevant by findings that the pre-TCR can bind with low affinity to MHC-peptide (Mallis et al, 2015; Li et al, 2021). The pre-TCR also forms an immunological synapse with many similarities to the immunological synapse formed by the αβTCR, oriented towards MHC (Allam et al, 2021). More work is needed to determine any functional connection between CD5 and tuning, or epigenetic modifications during β-selection, but these data point to a role for CD5 during the DN3bPre stage in reporting pre-TCR activity to determine subsequent passage through the β-selection checkpoint.

The molecular basis by which ACY1215 mediates the effects on cell fate we observed here needs more exploration. There are many possible means by which ACY1215 might impact upon transcriptional and epigenetic alterations during β-selection to mediate the phonotypes we observe here. At β-selection, transient activation of β-catenin mediates TCF-1 and Lef1 transcriptional regulation to promote pre-TCR–dependent transition to the DN4 stage (Xu et al, 2009). Ectopic expression of a constitutively active β-catenin allows the cells to bypass pre-TCR signalling, yielding DN4 cells lacking expression of TCRβ (Gounari et al, 2001). The similarities between this phenotype and the apparent bypass of pre-TCR signalling we observe with ACY1215 treatment suggest TCF-1/Lef might mediate the effects of ACY1215. Indeed, HDAC6 can regulate acetylation of β-catenin to coordinate its degradation, providing a possible mechanism for the effects of ACY1215 on TCF-1/Lef activity (Iaconelli et al, 2015). Given the correlation between Lef1 expression and CD5 we observe, CD5 might be downstream of Lef1 during β-selection, perhaps explaining the ACY1215-mediated deregulation of CD5

expression. These findings suggest that the induction of Lef1 at the DN3b^Pre stage is coordinated with pre-TCR signalling, in part to promote CD5 expression, and that this activity can be altered by ACY1215 treatment. Whether the effect of ACY1215 relates to HDAC6, an alternative HDAC or the histone deacetylase activity of Lef1/TCF-1 remains to be determined.

Other molecular mediators downstream of ACY1215 effects are also possible. It was recently shown that the transcriptional repressor Bcl6 is induced by pre-TCR signalling and required for passage through the β-selection checkpoint (Solanki et al, 2020). Differentiation of T follicular helper cells also requires Bcl6, in this case, induced by Lef1 and TCF (Choi et al, 2015). Several HDACs have been shown to complex with and modulate Bcl6 activity to impact on B cell development and function (Wang et al, 2020). Together with our findings, these studies suggest the possibility that ACY1215 might mediate at least some of its effects via Bcl6. HDAC6 has also been implicated in the DNA damage response (Zhang et al, 2019), raising the possibility that ACY1215 effects might relate to the genetic recombination integral to the creation of a TCRβ chain (Miyazaki et al, 2008). In any case, the functional outcome of ACY1215 treatment is to disrupt the connections between pre-TCR signalling and expression of CD5 expression and CD5-associated proliferation. Uncoupling CD5 from TCRβ and Lef1 expression provides a possible mechanism by which ACY1215 allows the wrong cells (perhaps cells without suitable TCRβ) to proliferate and differentiate into DN3b^Post cells, but these cells subsequently fail to progress, leading to the block in DP differentiation.

The known role of HDAC6 in linking tubulin acetylation to the immunological synapse of mature T cells (Serrador et al, 2004), and our finding that acetylation of α-tubulin is increased in DN3 cells suggested the possibility that the β-selection immunological synapse might mediate the effects of ACY125. Indeed, ACY1215 does alter the organisation of the TCR signalling molecules, LAT, around the MTOC. Similarly, Notch1, which is required to enhance the polarisation of the pre-TCR (Charnley et al, 2020; Allam et al, 2021), is lost from the MTOC after ACY1215 treatment. These findings are particularly interesting in light of a recent report that HDAC6 is required for the formation of aggresomes at the MTOC (Magupalli et al, 2020) and suggest that HDAC6 might play a more general role in coordinating recruitment to the MTOC. An alternative explanation for the effect of ACY1215 on β-selection relates to observations that HDAC inhibitors can induce MHC constituents to promote antigen presentation (Magner et al, 2000). It remains to be determined whether an influence on stromal-presented peptide MHC plays any role in disrupting β-selection. Interestingly, both CD2 and CD5 can regulate activity of the immunological synapse and connectivity to the MTOC (Dustin et al, 1998; Tibaldi et al, 2002; Brossard et al, 2003), further supporting the possibility that the immunological synapse provides a hub at which deacetylation and pre-TCR signalling converge. Together, these findings suggest that the effect of HDAC6 on β-selection that we see here might occur through disruption of the DN3 immunological synapse.

The molecular basis for the effects of ACY1215 on β-selection might therefore be multifaceted and relate to one or more of direct alteration of the epigenome to influence protein expression, alteration of Lef1 activity, DNA damage response, alteration of pre-

TCR signalling via organisation at the MTOC, and the immunological synapse. Irrespective of the molecular mechanism, the finding that AY1215 disrupts T cell development has potential implications for the development of clinical applications of small-molecule inhibitors of HDAC6, particularly in treatment of young patients whose T cell repertoire has not yet been established (Cosenza & Pozzi, 2018; Hogg et al, 2020).

# Materials and Methods

### Ex vivo culture of developing T cells and drug treatments

Two sources to derive developing T cells were employed in this study: mouse HSCs and mouse primary thymocytes. HSCs were collected from C57BL/6 mouse E14.5 fetal liver. To induce ex vivo T cell development, HSCs were co-cultured with OP9-DL1 stromal cells (from Juan Carlos Zúñiga-Pflücker, University of Toronto) at a 1:1 ratio in a 6-well plate ($2 \times 10^5$). The coculture was maintained in αMEM (M4526; Sigma-Aldrich), supplemented with fetal calf serum (10% vol/vol), glutamine (1 mM), β-mercaptoethanol (50 μM), sodium pyruvate (1 nM), HEPES (10 mM), penicillin/streptomycin (100 ng/ml), mouse interleukin 7 (1 ng/ml), and mouse Flt3 (5 ng/ml). C57BL/6 mice at 4–5 wk were euthanised for thymus collection. The primary thymocytes were released from the thymus for sorting, analysis, or further culturing with OP9-DL1 stromal cells under the same culture conditions as the HSC system described above. For both sources of developing T cells, ACY1215, purchased from Med-ChemExpress, was used at 1 μM as described (Laino et al, 2019). ACY1215 was dissolved in DMSO to 5 mM and diluted to 1 μM with culture media immediately before application to cells. The selective inhibitor of p300 and CBP, C646 (Sigma-Aldrich), was used at 2.5 μM. OP9-DL1 stromal cells were maintained in αMEM (M4526; Sigma-Aldrich), supplemented with 20% (vol/vol) fetal calf serum, glutamine (1 mM), and penicillin/streptomycin (100 ng/ml). When OP9-DL1 stromal cells were confluent after 3–4 d of coculture, primary thymocytes were dissociated from the stromal cells through forceful pipetting and were harvested with the fresh OP9-DL1 stromal cells.

### Flow cytometry and Western blot

To purify T cell populations using flow cytometry, primary thymocytes were dissociated from OP9-DL1 stromal cells and were resuspended in PBS buffer containing 2% FBS and antibody cocktail for surface staining. Antibody cocktails comprised lineage markers (NK-1.1, CD11b, CD45R, Ly6G6C, Ter119), and differentiation markers (CD4, CD8, CD25, CD44, CD28, and CD2; details about fluorochrome, manufacturer, and dilution are summarised in the Supplementary Materials). Primary thymocytes were incubated with antibody cocktails for 40 min before FACS.

Flow cytometry (BD Aria III) was used to analyse and sort the populations of developing T cells. DN3 cells were identified based on surface expression (Lineage⁻, CD4⁻, CD8⁻, CD25^Hi, CD44^Lo). DN3a cells were DN3 cells with CD28^Lo; by contrast, DN3b cells were DN3 cells with CD28^Hi. DP cells were identified based on surface expression (Lineage⁻, CD4⁺, CD8⁺).

To detect intracellular expression of TCRβ, Ki-67, Lef1, c-Myc, Notch1, HDAC6, and H3K18ac, the Foxp3/Transcription Factor Staining Buffer Kit (eBioscience) was used according to the manufacturer's instruction. Briefly, the primary thymocytes after surface staining were incubated with fixative for 15 min in dark, room temperature, followed with centrifuge and washing out the fixative. To detect expression of acetyl-tubulin, the cells were fixed with IC Fixation Buffer (eBioscience) for 15 min in dark, room temperature, followed with centrifuge and washing out the fixative. The fixed cells were then incubated with antibody for flow cytometry analysis.

To detect tubulin acetylation by Western blotting, cells were lysed in RIPA buffer, and lysates were subjected to SDS–PAGE and probed with antibodies to tubulin and acetylated tubulin.

### Immunofluorescence staining and confocal microscopy

The sorted DN3a or DN3b were grown on glass-bottomed 8-well chambers (ibidi) with OP9-DL1 stromal cells and were fixed with 4% formaldehyde for 8 min at room temperature followed by 0.1% Triton X-100 permeabilization for 5 min. The samples were then incubated with primary antibody overnight at 4°C (details about antibody usage are summarised in Table S1). After the primary antibody being washed out by PBS, the cells were labelled with fluorochrome-conjugated secondary antibodies (as Table S1), DAPI (Thermo Fisher Scientific), and phalloidin (Abcam). The samples were examined using an FV3000 confocal microscope (Olympus) and 60X lens (1.30 NA, UPLSAPO; Olympus). The images were processed with maximum-intensity projection using Image J. The mean intensity of region of interest was examined in Image J.

### Data collection and statistical analyses

Python (version 3.8.1) was used to visualize the statistics. Data of flow cytometry were analysed by FlowJo software (version X10.0.7r2; Tree Star, Inc.). The bar plot was shown as mean ± SEM. More than three repeats were included in the statistics and were colour-coded for each individual repeat. Microsoft Excel (version 2016) was used for two-sided $t$ test, by which $P$-value was indicated.

### Ethics

C57bl/6 mouse E14.5 fetal livers and 4–5-wk mouse thymi were collected according to the Peter MacCallum Cancer Centre Ethics Committee approval E627 and La Trobe University Animal Ethics Committee approval AEC-20024.

# Supplementary Information

# Acknowledgements

This work was done on Wurundjeri land of the Kulin nation, and we pay our respects to the Elders, past and present. We thank Louise Cheng, Callum Dark, Amr Allam, and Peter Fraser for technical assistance and helpful discussions and Ellis Reinherz, Jonathon Duke-Cohan for helpful comments on the article. The work was funded by Australian Research Council (FT0990405 to SM Russell), National Health and Medical Research Council (APP1099140 to SM Russell), and Swinburne University Postgraduate Research Award to AS Chann.

## Author Contributions

AS Chann: conceptualization, data curation, software, formal analysis, validation, investigation, visualization, methodology, and writing—original draft, review, and editing.
M Charnley: conceptualization, resources, methodology, and writing—review and editing.
LM Newton: resources and writing—review and editing.
A Newbold: resources and writing—review and editing.
F Wiede: investigation and writing—review and editing.
T Tiganis: resources.
PO Humbert: resources and writing—review and editing.
RW Johnstone: resources and writing—review and editing.
SM Russell: conceptualization, resources, data curation, software, formal analysis, supervision, funding acquisition, validation, visualization, methodology, project administration, and writing—original draft, review, and editing.

## Conflict of Interest Statement

The authors declare that they have no conflicts of interest.

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
