## [Reviewer comments · Life Science Alliance]

Life Science Alliance

Stepwise progression of β -selection during T cell development involves histone deacetylation

Anchi S Chann, Mirren Charnley, Lucas M. Newton, Andrea Newbold, Florian Wiede, Tony Tiganis, Patrick O Humbert, Ricky W Johnstone, Sarah M Russell
DOI: <https://doi.org/10.26508/lsa.202201645>

Corresponding author(s): Dr. Sarah M Russell (Peter MacCallum Cancer Centre)

Review Timeline:

Submission Date:	2022-08-03
Editorial Decision:	2022-09-02
Revision Received:	2022-09-26
Editorial Decision:	2022-09-27
Revision Received:	2022-10-02
Accepted:	2022-10-04

Transaction Report:

Please note that the manuscript was reviewed at Review Commons and these reports were taken into account in the decision-making process at Life Science Alliance.

Full Revision

Manuscript number: RC-2022-01414

Corresponding author(s): Russell, Sarah

[Please use this template only if the submitted manuscript should be considered by the affiliate journal as a full revision in response to the points raised by the reviewers.]

*If you wish to submit a preliminary revision with a revision plan, please use our "Revision Plan" template. **It is important to use the appropriate template to clearly inform the editors of your intentions.**]*

1. General Statements [optional]

This section is optional. Insert here any general statements you wish to make about the goal of the study or about the reviews.

Reviewers 1 & 3 make the very valid point that we do not have evidence for HDAC6 as the molecular target for the effects of ACY1215 on T cell development. We agree entirely, and do not claim to have defined the molecular target of ACY1215. However, these findings have direct relevance to the current use of ACY1215 as a cancer therapeutic. In addition, they provide a valuable new means of understanding the sequence of events in β -selection at higher resolution than was previously possible.

A second major aspect of concern was whether the drug acted upon T cell development directly, or through effects on OP9 stromal cells. We hope you will agree that our new data (detailed below) has put that concern to rest.

This section is mandatory. Please insert a point-by-point reply describing the revisions that were already carried out and included in the transferred manuscript.

Reviewer #1 (Evidence, reproducibility and clarity (Required)):

In this work, Russell and colleagues study the differentiation steps of thymocytes around the time of expression and function of the pre-TCR. They use CD2 as a new marker of an intermediate population of differentiation between DN3a thymocytes and DN4 thymocytes. The CD2+ thymocytes would be and a later stage of differentiation than the CD2 negative ones. In this way, they define a DN3b-Pre and a DN3b-Post populations. The study also aims to study the role of histone deacetylase HDAC6 in thymocytes differentiation at the time of pre-TCR signaling by using as a tool the inhibitor ACY1215 and conclude that HDAC plays an important role during the DN3-DN4 transition.

The method the authors use to study consist of the sorting of precursor cells from fetal or adult thymuses and studying the effect of the inhibitor during differentiation in vitro promoted by interaction of the thymocytes with the OP9-DL1 cell line, a well-characterized cell line that expresses a ligand of Notch and promotes DN3 differentiation up to the DP stage.

There are major concerns with the approach used by the researchers that make their conclusions unsustainable.

1) They use the inhibitor ACY1215 in the co-culture of thymocytes with OP9-DL1 cells to examine the effect of HDAC inhibition in thymocytes on their differentiation. However, the authors do not provide any control result showing that the effects detected on differentiation are not caused by the activity of the inhibitor on OP9-DL1 cells and not on thymocytes. This possibility is not excluded and if the inhibitor were acting on OP9 cells would invalidate completely the conclusions of the study.

We now include data that ACY1215 does not impact upon OP9-DL1 cell number or acetylation of tubulin, and that ACY1215-treated OP9-DL1 cells can fully support adhesion and differentiation of thymocytes (new Supp Fig 2 A-C). We have added two additional authors who contributed to these data. We believe these data provide strong evidence that ACY1215 exerts its effects directly on the developing T cells.

2) The inhibitor is used at a single concentration (one has to search in the Methods section in order to find that it is 1 micromolar) and there is no evidence provided of dose-response effects. Furthermore, experiments are missing in which different doses of the inhibitor are tested on its potential targets and shown to have a selective impact on those targets at the concentration used in the study.

We do not believe these analysis are within the scope of the manuscript, the focus of which is the novel differentiation characteristics revealed by ACY1215, rather than the impact of the drug itself.

3) The authors suggest that the inhibitor is a epigenetic regulator because it acts on the deacetylation of histones but also that it acts on the formation of a potential immunological synapse by the pre-TCR expressing thymocytes with inhibition of the translocation of the microtubule organizing centre. By the end of the study, one does not know exactly how is the inhibitor acting on pre-T cell differentiation.

We completely agree, this manuscript provides a starting point for dissecting the molecular mechanism for ACY1215 effects, and we have taken care to discuss the several possibilities that are currently in play. However, we believe the identification of an effect is a critical finding in its own right and has already proven valuable in uncovering novel events in β -selection.

4) The authors seem to postulate a model in which the effect of the inhibitor on the immunological synapse and pre-TCR signaling affects the expression of Lef1 which itself modulates histone deacetylation. Therefore, the inhibition of HDAC6 would be acting on pre-

TCR signaling upstream but also on the activity of Lef1 downstream. It seems that the approach is not sufficiently precise as to discriminate the site of action of the inhibitor

We propose this as one possible model. Dissecting the impact via Lef1 vs HDAC6 vs tubulin is beyond the scope of this paper.

5) The authors use contour plots to display their data. This has the advantage of defining cell populations but on the other hand, hides the number of events that are defining the populations. This is for instance reflected in Figure 7 panel B where the CD5 vs surface TCRbeta plot of cultured DN3a thymocytes shows many different populations (in green) which are probably artifacts of the contour plot. Very likely such populations are formed by very few events

We have adjusted the contour densities. We also note that the histograms above and to the side of the contour plots are provide for ready comparison of proportions.

6) I do not see a big effect of the inhibitor on Lef 1 expression (Figure 6A) and if the inhibitor has an effect at the DN3b-Pre stage why it should not have it at the DN3b-Post stage.

The effect of ACY1215 on Lef1 expression is clear in FL-derived DN3b^{Pre} (Fig 6Ai) and in thymus-derived DN3a and DN3b^{Pre} cells (Fig 6Aii,7A). Given previous findings that Lef1/TCF are required for progression through β -selection (Xu et al, 09 from the manuscript), these data suggest that the failure to upregulate Lef1 in some DN3b^{Pre} cells might prevent their traversal to DN4, explaining why DN3b^{Post} cells don't exhibit a loss of Lef1.

7) Suppl. Fig. 11.-In DN3b-Post, expression of CD5 is higher than in DN3a but not so clearly higher than in DN3b-Pre. The effect of the inhibitor on DN3b-Post was that of reducing CD5 expression but not Lef1 expression. In this experiment, the effect of the inhibitor of Lef1 expression by DN3b-Pre is not seen. Quantitation? The inhibitor could be altering CD5 expression by inhibiting the synapse independent of Lef1 ?

This is an understandable misconception, and we have clarified in the text and by including a plot of CD5 vs Lef1 in Sup Fig 11. The effect of drug on Lef1 expression in DN3b^{Pre} is readily apparent (see reduction in Lef1^{Hi} in the red contour plot LHS). However, by separating Lef1 high and low populations (RHS), we see Lef1^{Hi} cells in the pink contours. This does not indicate a high proportion of Lef1^{Hi} cells, merely that we enriched for them in the gating, as a means to assess any correlation with CD5. The reviewer is absolutely correct that the inhibitor could be altering CD5 expression by inhibiting the synapse independent of Lef1. However, the clear correlation of expression of Lef1 and CD5 in untreated cells, and the loss of that correlation after ACY1215 treatment, supports the notion that ACY1215 disrupts a functional association between expression of Lef1 and CD5.

8) *Figure 8.- The populations defined by CFSE staining are too broad in terms of fluorescence intensity as to determine number of cell divisions. It seems that there are only two populations: one that has not diluted CFSE and therefore has not divided and another that has diluted CFSE and has divided. How many times? we do not know. On the other hand, CD5 is increased in cells that have diluted CFSE. Have they expressed CD4, CD8, downregulated CD25, other markers indicating that the cells are not longer DN3a or DN3b? CD5 is upregulated after CFSE dilution not before dilution, so is CD5 expression cause or effect of differentiation..The effect of the inhibitor is not clear.*

We agree the CFSE staining does not indicate number of divisions. We don't agree that there are two populations, rather, a spread of CFSE indicating heterogeneity in the extent of proliferation, and a inverse correlation between CFSE and CD5 indicating that proliferation is associated with increasing CD5 expression. At this stage, we do not believe there is sufficient information to predict a causal relationship between these two markers. However, to our knowledge, the association is novel and provides a strong basis for further exploration, particularly in light of recent published findings that CD5 can act to tune the TCR signal at later stages of T cell development.

Referees cross-commenting

Totally in agreement! The effect of the drug could be on the OP9-DL1 cells and not on thymocytes. This is not proven

We hope you agree that our new data alleviates this concern, and supports a T cell autonomous impact of ACY1215.

Reviewer #1 (Significance (Required)):

With all the concerns about the method used by the authors to define subpopulations in DN3-DN4 transition and the involvement of HDAC6 activity I do not believe the paper will have a significant impact in the field. The authors will need to rethink their approaches in order to investigate the subject with sufficient guarantees. Perhaps using genetic approaches.

As agreed by Reviewers 2 and 3, new findings in the manuscript represent a substantial contribution to the field irrespective of the molecular target of ACY1215. Genetic approaches have their own drawbacks (including issues of compensation in the non-acute setting of most genetic modifications), and the clinical use of inhibitors such as ACY1215 mean these findings are significant irrespective of the molecular target.

Reviewer #2 (Evidence, reproducibility and clarity (Required)):

While the authors appear to have initially set out to determine the role of HDAC6 at the beta-selection checkpoint of T cell development, they also came away with a new panel of markers that further delineate important stages of thymocyte differentiation, describing the sequential upregulation of CD28, the preTCR, Lef1 and CD5, and, subsequently, CD2 as cells pass the test verifying successful recombination of the T cell receptor (TCR) b chain. The authors used several approaches for this study; they took advantage of the well-characterized OP9-DL1 co-culture system to support differentiation of progenitor cells in the presence or absence of an HDAC6-specific inhibitor or they isolated progenitor populations of interest directly from a murine thymus for short term culture in the presence or absence of inhibitor. They identify an important role for HDAC6 at the b-selection checkpoint as evidenced by an accumulation of a subset of 'double negative 3' cells and attribute this, in part, to increased acetylation of a-tubulin at the preTCR synapse as well as dysregulated expression of and/or recruitment to the microtubule-organising centre of proteins essential for beta-selection signals. Possible disruption of preTCR signaling by an HDAC6 inhibitor appears to limit upregulation of regulatory molecules (e.g. CD5) and normal progression through this stage.

I find the data presented and conclusions made to be largely convincing and that this manuscript represents a substantial contribution to the field. I recognize many of the challenges associated with this study that include assessing protein levels on small populations of cells, the heterogeneity of the cell populations (even if sorting a discrete subset, the requirement for culturing the cells +/- inhibitor invites differentiation), and the pleiotropic effects of HDAC inhibitors (HDACi). With few exceptions, we can accept these issues as the authors were clearly careful in their approaches, and there are limited other techniques that can be used to address some of these questions.

We thank the reviewer for their strong endorsement of the study.

Major comments.

Given the effort already put into this paper, it might be worthwhile to ultimately show step-wise progression through the developmental intermediates described in this manuscript. A time course to assess differentiation of isolated DN3a cells in OP9-DL1 coculture could be considered, for example. The authors clearly have the reagents and skills to carry out these experiments though the timing would be a factor in order to capture the appearance of discrete developmental intermediates. This would, in many ways, provide a straightforward summary of many of the reported results that would improve accessibility to a broader audience.

We agree that this would be of ultimate value. Unfortunately, it is not in the scope of this manuscript.

The differences in protein levels stated are not always so obvious based on the contour plots and/or histograms; these are often subtle effects that can be obscured by the 'noise' that accompanies the analysis of protein levels on small populations of cells. Quantification of the data should be included when making conclusions about differences in expression levels of different markers. In addition, when quantification and statistics are provided, it is clear that at least three repeats were included, but without quantification, the reproducibility of the experiments are not known; the number of replicates for each experiment should be obviously stated.

We have added quantification to Fig 6, Fig 7 and Fig S11 and included numbers to indicate number of replicates for each experiments.

Minor points.

The impact of the inhibitor seems to be inconsistent in Fig. 6A i and ii; there is not much of an impact of the inhibitor in ii as I understand. There also seem to be significantly different expression patterns of Lef in the DN3b^{Pre} stage in the two models (OP9 vs ex vivo) - what is the explanation for this?

Lef1 does appear to be upregulated slightly earlier in ex vivo thymocytes compared to fetal liver derived cells, although most of the upregulation occurs in the DN3b^{Pre} stage in both systems. We are not sure of the explanation for this. Importantly both systems show clear reduction of Lef1 before, but not after, CD2 expression.

That the inhibitor leads to a 'loss of correlation' between Lef and CD5 levels is more obvious in Supp Fig 11 than as presented in Figure 7.

Perhaps, although it is still evident in Figure 7. We believe the effect is most striking when thymocytes were sorted for DN3a, and then treated for 2 days (Fig 7B), but we also see the effect at 1 day on fetal liver-derived cells (Supp Fig 11) and at 1 day for thymocytes (Fig 7B)

Consider adding a CD25^{lo} population (e.g. DN4) as a reference in Supp. Fig. 12.

By definition DN3 and DN4 are delineated by expression of CD25, so all DN4 cells are lower for CD25 than any of the DN3 subsets. We have now made this clear with addition of DN4 in Supp Fig 12 as requested.

In reference to Fig. 5, the authors suggest the analysis of MTOC components in DN3a cells; are these Dn3a cells? DN3a cells were isolated and then cultured for a day +/- inhibitor. I presume there is some differentiation in these cultures that depends on the presence of the inhibitor. I am not suggesting that the authors need to redo this experiment but rather either acknowledge that these are not all DN3a cells (unless I am wrong here) by changing the wording, or add a marker to distinguish the subset (CD2?)

Full Revision

You are correct, we have changed the wording to say: DN3a cells were cultured for 1 day with and without ACY1215 (so would be predominantly DN3a and DN3b^{Pre} and not yet past β -selection; see Supp Fig 9).

Make note of the DN3b nomenclature used; I believe that 'pro' was used instead of 'post' at least once.

We apologize, and have corrected.

In Fig 1, the populations identified as CD4+ and CD8+ are likely immature populations; unless including TCRb staining to distinguish these, I suggest excluding them from the analysis. I think the DP population is sufficient here to get the point across.

We appreciate the advice and have removed the SP data.

In Supp Fig 2. the schematic does not appear to be consistent with legend (2 versus 4 d of culture).

We have corrected to 2D.

In Supp Fig 8a, the quantification of DN3a/b-pre/b-pro has a different experimental set up than in Fig 3bii but is written as if this is the quantification of the data as I understand it.

No, the quantification is for Fig 3bi.

In line 214/215, it is suggested that "Analysis of the cells immediately after extraction showed clearly distinct DN3a, DN3bPre and DN3bPost cells." but I did not find the data for this.

We have now made more clear that this data is at the top right of Fig 4)

Please confirm the experimental set up in Supp. Fig. 9. Why treat with inhibitor prior to isolating the subsets and then culturing again without inhibitor?

This is the correct set-up. The goal was to enrich sufficient DN3bPre cells to ensure a pure sort (taking advantage of the ACY1215 effect), and then to monitor their differentiation.

Eliminate conclusions made without specific reference to figures or to 'future' figures.

We have tried to find such conclusions, but not been able to.

In some histograms (e.g. 5), it was not obvious to me what the negative controls represented.

Full Revision

Are these fluorescence minus one, isotype, other?

We have made more clear in the Figure Legends.

In the abstract, it is suggested that increases in a number of markers provides for escalating TCR signaling strength; CD5 is among these. As a negative regulator of TCR signals, this statement seems to be counterintuitive.

Apologies for this error, we have changed 'escalating' to 'modulating'

The authors state that, "These data together indicate that CD5 serves as a link between TCRb expression and proliferation," (line 325/326); how CD5 'links' the two is not clear.

We have changed the wording.

-It is written that "...at Day 8-10 of the co-culture, when cells from mouse fetal liver were predominantly at the DN3 stage of T cell development (Supp Fig 1B)...". Reconsider wording this statement as it appears as if the majority of cells actually express CD4 and/or CD8.

Thank you, we have done so.

****Referees cross-commenting****

Reviewers 1 and 3 bring up important, obvious points that this reviewer missed in terms of the potential off-target effects of the HDACi on the stromal cell component of the co-culture system used for the experiments. It is difficult to determine the relevance of HDAC6 at this developmental checkpoint without additional controls

We hope you agree that we have allayed this concern with new controls.

Reviewer #2 (Significance (Required)):

As an immunologist with an interest in the molecular and cellular mechanisms that regulate T cell development, I find this manuscript interesting for several reasons.

One of the benefits of studying development and differentiation in the immune system is the ability to distinguish discrete developmental intermediates by flow cytometry using defined panels of cell surface and intracellular markers; this allows for the isolation of populations of cells for testing progenitor/progeny relationships, to identify the role of essential genes at

various developmental stages and beyond. This manuscript adds important new markers that will allow researchers to more discretely tease apart important stages in T cell development during an important regulatory checkpoint.

Not only is beta-selection an essential first step in ensuring a functional antigen receptor repertoire during T cell development, but its tight regulation is absolutely necessary due to the double-strand DNA breaks that accompany antigen receptor gene rearrangement and the massive proliferation that ensues after successful pairing of a functionally rearranged TCR β chain with the preTCR α ; indeed, dysregulation at the beta-selection checkpoint can give rise to leukemia. This study provides new insight into potential mechanisms of beta-selection regulation.

Thank you for this positive evaluation.

Reviewer #3 (Evidence, reproducibility and clarity (Required)):

Chann et al employ a pharmacologic inhibitor (ACY1215) to assess the role of HDAC6 as a molecular effector of differentiation during traversal of the b-selection checkpoint. Using this inhibitor, they detect an accumulation of cells at the DN3b stage that have failed to upregulate CD2. Consequently, they propose the existence of a new intermediate stage between induction of CD28 and CD2 and term these stages as DN3b-pre and DN3b-post. The proposal is that the arrest of development between these two stages results from interference with the normal pattern of induction of CD5 and Lef1. The experiments are thoughtfully designed and interpreted. However, there are a number of significant issues.

1. HDAC6 ko mice have no thymic phenotype (Zhang MCB 08; Fig. 6) raising the possibility that all of the effects observed following ACY1215 treatment result from off-target activity. This calls the mechanistic analysis into question. One way to address this would be to treat HDAC6 deficient thymic progenitors with equivalent doses of drug to determine if the observed effects do not occur.

We agree that HDAC6 might not be the primary target, and have gone to considerable lengths to make this clear in the manuscript. Rather than focus specifically on HDAC6, we believe identification of all possible molecular targets (HDAC6, Lef1/TCF1, perhaps other acetylases, each conferring transcriptional or cytoskeletal regulation) will require extensive efforts not within the scope of this manuscript.

2. Equally important is that the analysis is essentially all descriptive with no interventions (gain or loss-of-function) to investigate the causal relationships of the correlations observed.

We don't agree that the analysis is essentially all descriptive, since our findings derived from the application of ACY1215.

3. Some of the data interpretation is puzzling. For example, drug treatment results in an increased proportion of DN3b cells, which is interpreted to mean that drug promotes the DN3a to DN3b transition; however, based on the absolute counts, a more likely explanation is the the drug is killing the DN3a cells (Fig2a).

We certainly agree and state that DN3a cells are depleted by 1 day ACY1215, but do not believe this is due to death given the apoptosis analysis in Fig S4. Given that DN3 and DN4 cell numbers are equivalent even out to Day 4 when one would expect the DN3a depletion to have substantial effect, we suggest that this is 'perhaps caused by precocious differentiation from DN3a to DN3b'.

Referees cross-commenting

Is there consensus that w/o clear evidence pointing to on-target action of the drug on the intended target that the significance of the study is limited?

We have confirmed that thymocytes are the cellular target. The molecular target will take much more work and is not the focus of this paper.

Reviewer #3 (Significance (Required)):

Because of the complete absence of a thymic phenotype of HDAC6-deficient mice, the significance of these findings is substantially in doubt.

There are many examples of key biological processes that were not revealed by a phenotype in the knockout, due to compensatory effects (pertinent to this paper: Lef1 being a clear example).

September 2, 2022

Re: Life Science Alliance manuscript #LSA-2022-01645

Dr. Sarah M Russell
Peter MacCallum Cancer Centre
Immune Signalling Laboratory
St. Andrew's Place
East Melbourne, Victoria 3002

Dear Dr. Russell,

Thank you for submitting your revised manuscript entitled "Stepwise progression of beta-selection during T cell development as revealed by histone deacetylation inhibition" to Life Science Alliance. The manuscript has been seen by the original reviewers whose comments are appended below. While the reviewers continue to be overall positive about the work in terms of its suitability for Life Science Alliance, some important issues remain. Please address the remaining Reviewer 1 issues.

Our general policy is that papers are considered through only one revision cycle; however, given that the suggested changes are relatively minor, we are open to one additional short round of revision. Please note that I will expect to make a final decision without additional reviewer input upon resubmission.

Please submit the final revision within one month, along with a letter that includes a point by point response to the remaining reviewer comments.

To upload the revised version of your manuscript, please log in to your account: <https://lsa.msubmit.net/cgi-bin/main.plex>
You will be guided to complete the submission of your revised manuscript and to fill in all necessary information.

B. MANUSCRIPT ORGANIZATION AND FORMATTING:

Sincerely,

Reviewer #1 (Comments to the Authors (Required)):

The work is commendable, and the data generally support the conclusions made by the authors in terms of differential

expression of markers that indicate discrete stages of b-selection. The study, itself, is very complicated and accessibility to a broader audience might be limited.

Remaining issues to be addressed:

- There was a request for quantification of data relevant to differences in expression levels and clarity in the number of replicates in each experiment. The authors suggest that "We have added quantification to Fig 6, Fig 7 and Fig S11 and included numbers to indicate number of replicates for each experiments." It is not clear to me that this is complete; are the authors suggesting the all the quantification of data in Fig 6 and 7 is in S11? In addition, there are other figures in which the number of replicates is not clearly stated as far as I can tell (e.g. Fig. 8).

- Quantification of CD4 and CD8 "SP" populations were removed from Fig. 1, but new panels were added that analyze these populations (e.g. Supp Fig 5A). These are very likely not mature thymocyte populations and should not be identified as such without additional markers.

Issues that were sufficiently, but not ideally, addressed:

- This is a complicated study; to be more broadly accessible and ultimately show stepwise progression through the developmental intermediates associated with the b-selection checkpoint, a time course to summarize markers associated with the differentiation of isolated DN3a cells in OP9-DI1 coculture (or similar experiment) would have been ideal. I appreciate that the authors summarize their findings of the temporal modulation of markers of differentiation and provide a working model (also in the original manuscript) for the data.

- The authors performed several experiments to show that the b-selection block in the co-culture system was due to the HDACi acting on the thymocytes and not the stromal cells. They show that the HDACi does not impact α -tubulin acetylation in OP9-DL1 cells, that treatment of the stromal cells with the inhibitor does not disrupt thymocyte attachment, and that no overt block in differentiation is observed when thymocytes are cultured on OP9-DL1 cells that had recent exposure to the inhibitor.

While the most straightforward test might be to culture WT vs HDAC6-deficient thymocytes on OP9-DL1 cells +/- inhibitor as another reviewer suggested; the new experiments largely address reviewer concerns. Ideally, a positive control for HDACi activity (e.g. H3K18 acetylation in the stromal cells?) and concurrent co-cultures of OP9-DL1 and thymocytes with or without the inhibitor would have been included (as I'm not sure this exact experiment - starting population to end read out - was performed elsewhere in the manuscript for a direct comparison). Ultimately, there was clear effort made to address the reviewers' issues. In my opinion, this is largely sufficient, though I defer to my reviewer colleagues who originally marked this as a major concern.

Reviewer #2 (Comments to the Authors (Required)):

The study by Chann et al remains a largely descriptive study involving treatment with a pharmacologic agent (ACY1215), for which there is no clear target. Effects on thymocyte development are described that purports to define a new developmental intermediate. The authors make the case that it is not their aim to identify the drug target, which is not trivial, but their choice not to establish or investigate this diminishes impact. Moreover, despite the careful analysis of events resulting from drug treatment, their analysis of the effects of drug treatment remain largely descriptive. There are no gain/loss-of-function experiments to determine if the changes in target expression (CD5, Lef1, etc) or events are causally linked to the observed perturbations in development or merely correlated. This also substantially diminishes the impact of this study. Nevertheless, this collection of observations is of value to the community and they should be allowed to judge it on its own merits.

Dear Dr Sawey,

Thank you for your positive review of our manuscript "Stepwise progression of beta-selection during T cell development as revealed by histone deacetylation inhibition".

Below is a point-by-point response describing how we have addressed the remaining Reviewer 1 issues.

Reviewer #1 (Comments to the Authors (Required)):

The work is commendable, and the data generally support the conclusions made by the authors in terms of differential expression of markers that indicate discrete stages of b-selection. The study, itself, is very complicated and accessibility to a broader audience might be limited.

Remaining issues to be addressed:

-There was a request for quantification of data relevant to differences in expression levels and clarity in the number of replicates in each experiment. The authors suggest that "We have added quantification to Fig 6, Fig 7 and Fig S11 and included numbers to indicate number of replicates for each experiments." It is not clear to me that this is complete; are the authors suggesting the all the quantification of data in Fig 6 and 7 is in S11? In addition, there are other figures in which the number of replicates is not clearly stated as far as I can tell (e.g. Fig. 8).

We apologize for not providing more clarity. Because the multiplexed flow cytometry data was presented with iterative addition of new markers once they became relevant to the story, some of the quantification was presented in earlier figures. We have now made that more clear in the text, and have both moved the quantification from the previous supplementary Fig 11 (ii) to main figures (eg the current Fig 7A) and clarified where the quantification is presented in the supplementary figures. The reviewer is correct that some quantification was missed, and we have now added that quantification in Fig 7(B) and described experiment numbers throughout. To accommodate these new panels, we have slightly rearranged the last two figures – moving some quantification to the main Fig 7 (as said above), and moving the schematic of the previous Fig 7 down to the current Fig 8 to have the two schematic models all in the same figure, and moving the data in the previous Fig 8 to the current supplementary Fig 14B. We hope you will agree that these changes both complete the statistical analysis, and make the paper easier to follow.

- Quantification of CD4 and CD8 "SP" populations were removed from Fig. 1, but new panels were added that analyze these populations (e.g. Supp Fig 5A). These are very likely not mature thymocyte populations and should not be identified as such without additional markers.

Apologies, we have now removed the panels in Fig 5a and Supp Fig 5A, and confirmed that SP data is not included anywhere else in the manuscript.

Issues that were sufficiently, but not ideally, addressed:

- This is a complicated study; to be more broadly accessible and ultimately show stepwise progression through the developmental intermediates associated with the b-selection checkpoint, a time course to summarize markers associated with the differentiation of isolated DN3a cells in OP9-DL1 coculture (or similar experiment) would have been ideal. I appreciate that the authors summarize their findings of the temporal modulation of markers of differentiation and provide a working model (also in the original manuscript) for the data.

We agree this would be desirable, but unfortunately not feasible.

- The authors performed several experiments to show that the b-selection block in the co-culture system was due to the HDACi acting on the thymocytes and not the stromal cells. They show that the HDACi does not impact α -tubulin acetylation in OP9-DL1 cells, that treatment of the stromal cells with the inhibitor does not disrupt thymocyte attachment, and that no overt block in differentiation is observed when thymocytes are cultured on OP9-DL1 cells that had recent exposure to the inhibitor.

While the most straightforward test might be to culture WT vs HDAC6-deficient thymocytes on OP9-DL1 cells +/- inhibitor as another reviewer suggested; the new experiments largely address reviewer concerns. Ideally, a positive control for HDACi activity (e.g. H3K18 acetylation in the stromal cells?) and concurrent co-cultures of OP9-DL1 and thymocytes with or without the inhibitor would have been included (as I'm not sure this exact experiment - starting population to end read out - was performed elsewhere in the manuscript for a direct comparison). Ultimately, there was clear effort made to address the reviewers' issues. In my opinion, this is largely sufficient, though I defer to my reviewer colleagues who originally marked this as a major concern.

We are glad that the reviewer felt this was largely sufficient, and that Reviewer 2 raised no concerns.

We have made one additional change beyond those suggested by the reviewers – unified the presentation of T test statistics, which had been partly presented as $p < xxx$, and partly with asterisks. Now all explicitly state the p value range. We hope this improvement makes the paper a little easier to read.

We hope that these changes render the manuscript acceptable for publication in Life Science Alliance.

Sincerely,

Sarah Russell

September 27, 2022

RE: Life Science Alliance Manuscript #LSA-2022-01645R

Dr. Sarah M Russell
Peter MacCallum Cancer Centre
Immune Signalling Laboratory
St. Andrew's Place
East Melbourne, Victoria 3002

Dear Dr. Russell,

Thank you for submitting your revised manuscript entitled "Stepwise progression of β -selection during T cell development involves histone deacetylation". We would be happy to publish your paper in Life Science Alliance pending final revisions necessary to meet our formatting guidelines. Please revise and format the manuscript and upload materials by Thursday.

- please upload your main and supplementary figures as single files
- please use the [10 author names, et al.] format in your references (i.e. limit the author names to the first 10)

A. FINAL FILES:

B. MANUSCRIPT ORGANIZATION AND FORMATTING:

****It is Life Science Alliance policy that if requested, original data images must be made available to the editors. Failure to provide original images upon request will result in unavoidable delays in publication. Please ensure that you have access to all original**

data images prior to final submission.**

The license to publish form must be signed before your manuscript can be sent to production. A link to the electronic license to publish form will be sent to the corresponding author only. Please take a moment to check your funder requirements.

Thank you for your attention to these final processing requirements. Please revise and format the manuscript and upload materials by Thursday.

Sincerely,

October 4, 2022

RE: Life Science Alliance Manuscript #LSA-2022-01645RR

Dr. Sarah M Russell
Peter MacCallum Cancer Centre
Immune Signalling Laboratory
St. Andrew's Place
East Melbourne, Victoria 3002

Dear Dr. Russell,

Thank you for submitting your Research Article entitled "Stepwise progression of β -selection during T cell development involves histone deacetylation". It is a pleasure to let you know that your manuscript is now accepted for publication in Life Science Alliance. Congratulations on this interesting work.

DISTRIBUTION OF MATERIALS:

Again, congratulations on a very nice paper. I hope you found the review process to be constructive and are pleased with how the manuscript was handled editorially. We look forward to future exciting submissions from your lab.

Sincerely,
